# Ventromedial hypothalamic neurons control a defensive emotion state

Prabhat S Kunwar[1], Moriel Zelikowsky[1], Ryan Remedios[1], Haijiang Cai[1], Melis Yilmaz[1], Markus Meister[1], David J Anderson[1,2]*

[1]Division of Biology and Biological Engineering, California Institute of Technology, Pasadena, United States; [2]Howard Hughes Medical Institute, California Institute of Technology, Pasadena, United States

**Abstract** Defensive behaviors reflect underlying emotion states, such as fear. The hypothalamus plays a role in such behaviors, but prevailing textbook views depict it as an effector of upstream emotion centers, such as the amygdala, rather than as an emotion center itself. We used optogenetic manipulations to probe the function of a specific hypothalamic cell type that mediates innate defensive responses. These neurons are sufficient to drive multiple defensive actions, and required for defensive behaviors in diverse contexts. The behavioral consequences of activating these neurons, moreover, exhibit properties characteristic of emotion states in general, including scalability, (negative) valence, generalization and persistence. Importantly, these neurons can also condition learned defensive behavior, further refuting long-standing claims that the hypothalamus is unable to support emotional learning and therefore is not an emotion center. These data indicate that the hypothalamus plays an integral role to instantiate emotion states, and is not simply a passive effector of upstream emotion centers.

*For correspondence: wuwei@caltech.edu

**Competing interests:** The authors declare that no competing interests exist.

## Introduction

Across the animal kingdom, appropriate defensive behavior is key to survival. Accordingly, it is not surprising that the brain has evolved multiple circuits to control such behaviors. For example, studies in rodents have led to the conclusion that there are multiple, anatomically distinct pathways controlling learned (conditioned) and innate defensive responses (reviewed in [*Rosen, 2004*; *LeDoux, 2012*] but see [*Cezario et al., 2008*; *Martinez et al., 2011*; *Gross and Canteras, 2012*]). In these pathways, sensory inputs converge on the amygdala, whose output is relayed by the hypothalamus to downstream structures that control the defensive response. However, different subdivisions of the amygdala and hypothalamus are thought to control learned vs innate responses, in a parallel manner (*Figure 1A*; reviewed in [*Davis, 1992*; *Fanselow, 1994*; *LeDoux, 1995*, *2000*; *Gallagher and Chiba, 1996*; *Fanselow and LeDoux, 1999*; *Canteras, 2002*; *Rosen, 2004*; *Swanson, 2005*; *Gross and Canteras, 2012*; *LeDoux, 2012*; *Saper and Lowell, 2014*]).

In mammals, at least, defensive behaviors reflect internal emotion states (*Darwin, 1872*), which are subjectively perceived by humans as 'fear' or 'anxiety' (*Adolphs, 2010*, *2013*; *LeDoux, 2012*). A large body of evidence has established the amygdala, principally the lateral (LA), basolateral (BLA) and central (CEA) subdivisions (*Figure 1A*), as a brain region that plays a central role in the implementation of emotion states, based on its involvement in conditioned fear (reviewed in [*Davis, 1992*; *Fanselow, 1994*; *LeDoux, 1995*, *1996*, *2000*]); for convenience we will henceforth use the shorthand term 'emotion center' to refer to such regions. In contrast, the hypothalamus is viewed primarily as a relay between the output of the amygdala, and downstream structures that generate observable behavioral, autonomic and endocrine components of a conditioned defensive response (*Davis, 1992*; *LeDoux, 1995*, *2000*, *2012*; *LeDoux and Damasio, 2013*).

**eLife digest** Animals have evolved a large number of 'defensive behaviors' to deal with the threat of predators. Examples include reptiles camouflaging themselves to avoid discovery, fish and birds swarming to confuse predators, insects releasing toxic chemicals, and humans readying themselves to fight or flee.

In mammals, defensive behaviors are thought to be mediated by a region of the brain called the amygdala. This structure, which is known as the brain's 'emotion center', receives and processes information from the senses about impending threats. It then sends instructions on how to deal with these threats to other regions of the brain including the hypothalamus, which pass them on to the brain regions that control the behavioral, endocrine and involuntary responses of the mammal.

For many years it has been thought that the role of the hypothalamus is to serve simply as a relay for emotion states encoded in the amygdala, rather than as an emotion center itself. However, Kunwar et al. have now challenged this assumption with the aid of a technique called optogenetics, in which light is used to activate specific populations of genetically labeled neurons. When light was used to directly activate neurons within the ventromedial hypothalamus in awake mice, the animals instantly froze and/or fled, just as they would when faced with a predator. Given that the optical stimulation had completely bypassed the amygdala, this suggested that the hypothalamus must be capable of generating this defensive response without any input from the amygdala.

The freezing and fleeing responses resembled the responses to a predator in a number of key ways. Mice chose to avoid areas of their cage in which they had received the stimulation, suggesting that—like a predator—these areas induced an unpleasant emotional state, perhaps akin to anxiety or fear. Freezing and fleeing persisted for several seconds after the stimulation had stopped, just as freezing and fleeing responses to predators do not immediately cease after the threat has gone. And finally, destroying the neurons targeted by the stimulation made mice less likely to avoid one of their main predators, the rat. It also made the animals less anxious.

Overall the results suggest that the hypothalamus may be more than simply a relay for the amygdala, and that 'amygdala-centric' views of emotion processing may need to be re-visited.

Because the identification of emotion centers has been rooted in their ability to mediate emotional learning (*LeDoux, 1996*; *Panksepp, 1998*, *2011b*), it has been challenging to ascertain whether innate defensive behaviors also reflect underlying emotion states, and therefore to investigate whether structures that mediate these behaviors, such as the medial hypothalamus (*Figure 1A*; reviewed in [*Canteras, 2002*; *Rosen, 2004*; *Swanson, 2005*; *Sternson, 2013*; *Saper and Lowell, 2014*]) serve as emotion centers. Indeed, classical studies reporting that electrical stimulation of the hypothalamus is unable to condition learned defensive responses (*Masserman, 1941*; *Wada and Matsuda, 1970*) have been used as evidence that the hypothalamus is not itself an emotion center, a view reflected in contemporary textbooks (*LeDoux and Damasio, 2013*). Instead, it has been assumed, by analogy to circuits mediating conditioned defensive responses, that emotion centers for innate defensive behaviors would be located in the medial amygdala (MEA), and that downstream hypothalamic targets would similarly serve as passive relays for amygdala output (*Gross and Canteras, 2012*; *LeDoux, 2012*; *LeDoux and Damasio, 2013*). Experimental testing of this assumption, however, has been hindered by the lack of more general criteria to identify and study emotion states in systems mediating unlearned defensive responses.

We have recently proposed that emotion states have several key properties that generalize across emotions and species. These properties include scalability (the magnitude and/or nature of the behavioral response varies with the level of arousal or intensity of the associated internal state), valence (positive or negative), generalization (a given state can be induced by multiple stimuli, and can control multiple behavioral responses) and persistence: they endure long after a threat is no longer present (*Russell, 2003*; *Posner et al., 2005*; *Anderson and Adolphs, 2014*). Furthermore we argue, in line with Darwin (*Darwin, 1872*), Cannon (*Cannon, 1927*) and others (*Panksepp, 1998*, *2011b*), that these emotion states play a causative role in controlling behavior (*Anderson and Adolphs, 2014*). If one accepts this premise, then behaviors that exhibit the general properties described above can be taken as evidence of an underlying emotion state with similar properties.

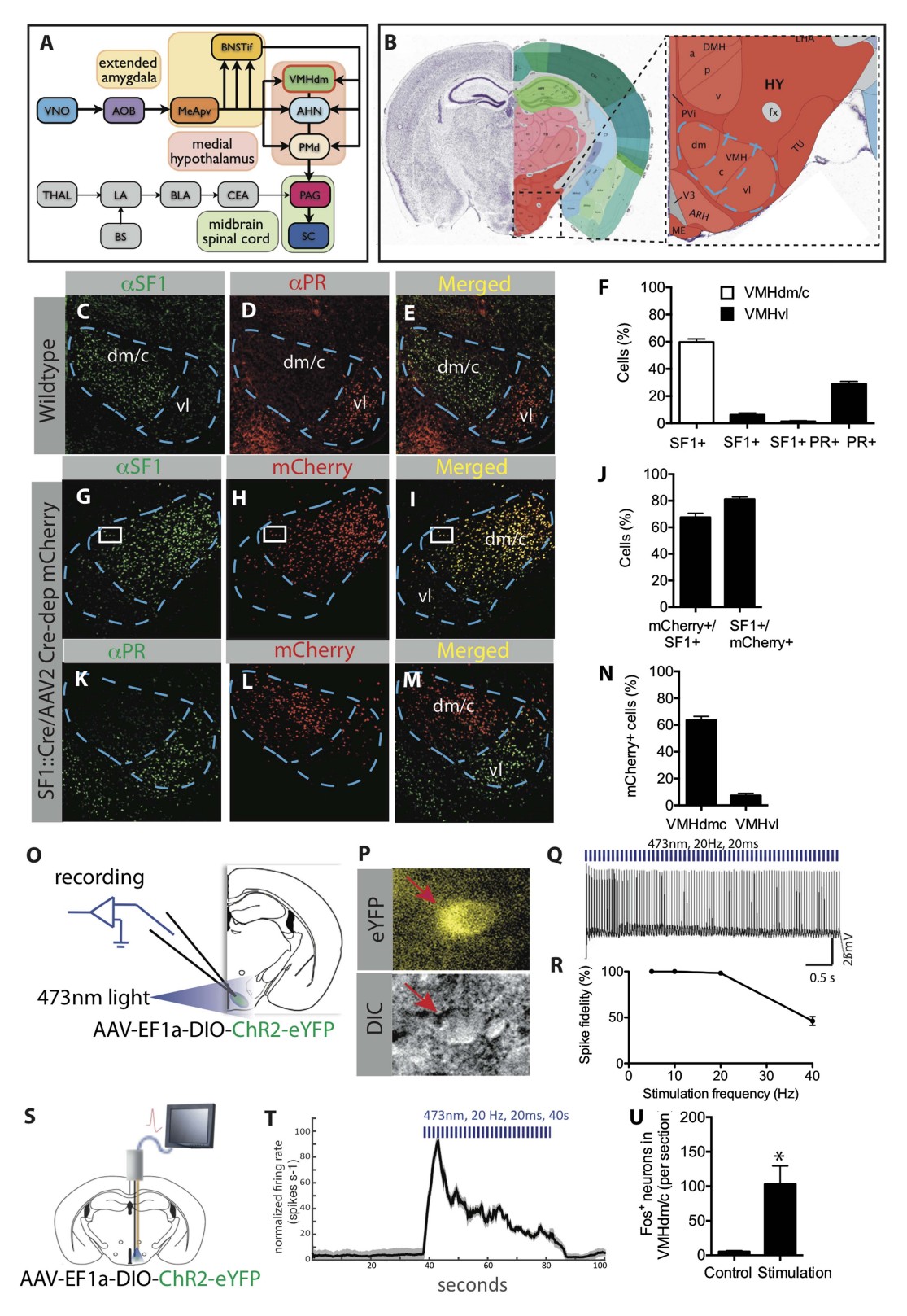

**Figure 1.** Characterization of SF1+ neurons and their optogenetic activation. (**A**) Schematic illustrating brain circuits involved in defensive behaviors. (**B**) Coronal section of the mouse brain showing the location of VMHdm (Allen Brain Atlas). VMH is indicated by the blue outline. (**C–E**) Representative images of the VMH in a wild type mouse showing double-label immunostaining for SF1 (green) and progesterone receptor (PR), a marker of VMHvl

*Figure 1. continued on next page*

*Figure 1. Continued*

neurons involved in social behaviors (red). (**F**) Percentage of cells in VMHdm/c (white bars) and VMHvl (black bars) that are SF1⁺ or PR⁺. n = 3 animals for each condition. (**G–I**) Representative images of VMH from an SF1-Cre transgenic mouse injected in VMH with a Cre-dependent AAV encoding mCherry (red) and immunolabeled with anti-SF1 antibody (green). (**J**) Percentage of overlap between VMHdm/c SF1⁺ cells and mCherry. n = 3 animals for each condition. (**K–M**) Representative images of VMH as in (**G–I**), double labeled for mCherry⁺ (red) and PR⁺ cells (green). (**N**) Percentage of total neurons that are mCherry⁺ in VMHdm/c and VMHvl (defined by domain of PR expression). n = 3 animals for each condition. (**O**) Schematic illustrating preparation for whole-cell patch clamp recordings of SF1⁺ neurons. (**P**) Representative photomicrograph of ChR2-eYFP-expressing (SF1⁺) cells (red arrow) patched for recording; DIC, differential interference contrast. (**Q**) Photostimulation-evoked spiking in neurons recorded as in (**P**). (**R**) Percent spike fidelity in (**Q**). n = 7 cells. (**S**) Schematic for in vivo electrophysiological response recordings from VMHdm/c in mice expressing ChR2 in SF1⁺ neurons. (**T**) Time-course of mean firing rate change in vivo in response to photostimulation. n = 6 units. (**U**) Average number of Fos⁺ neurons per section of VMHdm/c from photostimulated mice expressing ChR2 in SF1⁺ neurons. Control non-stimulated contralateral side within each animal. n = 4 animals for each condition. Values are represented as mean ± SEM. *p < 0.05.

The following figure supplement is available for figure 1:

**Figure supplement 1**. Projection profile of SF1⁺ (Nr5a1⁺) VMHdm/c neurons.

This broader and more general view of emotion states (*Anderson and Adolphs, 2014*), together with the availability of genetically based tools for cell type-specific manipulation of neuronal function (*Luo et al., 2008*; *Yizhar et al., 2011*; *Tye and Deisseroth, 2012*), provides an opportunity to revisit the role of hypothalamic neurons in controlling emotion states. The application of such tools in turn requires the identification of molecular markers for the cell types of interest. Recently, *Silva et al. (2013)* reported that pharmacogenetic silencing of neurons in the ventromedial hypothalamus, dorsomedial/central region (VMHdm/c; *Figure 1B*), which express the nuclear co-receptor Nr5a1 (also called SF1) (*Dhillon et al., 2006*), caused a reduction in defensive responses to a predator, but not to other types of threats, such as an aggressive conspecific or a footshock (*Silva et al., 2013*).

Here we have used time-resolved optogenetic gain-of-function manipulations of SF1⁺ neurons, as well as cell-specific ablation, to investigate their role in defensive behaviors and associated emotion states. We demonstrate that direct activation of these neurons, a manipulation that anatomically bypasses amygdala input, is sufficient to evoke multiple defensive behaviors, whose collective properties are consistent with the induction of an underlying defensive emotion state (*Anderson and Adolphs, 2014*). Ablation of SF1⁺ neurons, moreover, attenuates defensive behaviors in a variety of contexts. Finally, we show that SF1⁺ neurons can condition learned defensive responses to initially neutral contextual cues, further refuting earlier claims to the contrary (*Masserman, 1941*; *Wada and Matsuda, 1970*). Together these findings suggest that SF1⁺ neurons contribute directly and causally to a defensive internal emotion state.

## Results

### Characterization of VMHdm-specific SF1⁺ neurons

Neurons that express the gene *Nuclear receptor subfamily 5, group a* (*Nr5a1*), which is also referred to as *Steroidogenic factor 1* (*SF1*) (*Dhillon et al., 2006*; *Silva et al., 2013*) constitute about 60% of cells in VMHdm/c, and are not found in other hypothalamic or amygdalar regions (www.brain-map.org, *Nr5a1* in situ hybridization data). Double-labeling indicated that these neurons are essentially non-overlapping with subjacent VMHvl neurons mediating social behaviors such as aggression (*Yang et al., 2013*; *Lee et al., 2014*) (*Figure 1C–F*). To express different effectors in these neurons, we obtained a BAC transgenic mouse line that expresses *Cre*-recombinase under SF1 regulatory elements (SF1-Cre) (*Dhillon et al., 2006*). We validated Cre-specific recombination in SF1⁺ neurons by injecting stereotaxically, into the VMHdm/c of SF1-Cre mice, an adeno-associated virus (AAV) expressing a Cre-dependent mCherry reporter, and double labeling with an anti-SF1 antibody (*Figure 1G–J*). This analysis indicated that 80% of mCherry-expressing neurons were SF1⁺, while little expression was observed in VMHvl (*Figure 1K–N*).

In order to optogenetically manipulate SF1⁺ cells in VMHdm/c, SF1-Cre transgenic mice were infected with a Cre-dependent adeno-associated virus 2 (AAV2) containing an EF1α promoter-driven

channelrhodopsin-2 (ChR2 H134R) fused to eYFP (AAV-DIO-ChR2-eYFP) (*Boyden et al., 2005*; *Aravanis et al., 2007*). We characterized the physiological response of SF1[+] neurons to optogenetic stimulation using patch clamp recordings in acute VMH slices (*Figure 1O–R*), as well as by in vivo extracellular recordings (*Figure 1S–T*). In both cases, time-locked spiking was evoked by photostimulation, with 100% spike fidelity maintained up to a stimulation frequency of 20 Hz (*Figure 1Q,R*). Additionally, we observed strong Fos induction in the VMHdm/c region following photostimulation, providing further evidence of activation in vivo (*Figure 1U*).

## Activation of SF1[+] neurons induces freezing or activity bursts, depending on the level of photostimulation

When threatened, animals will display species-specific defensive reactions (*Bolles, 1970*; *Blanchard et al., 1990*, *1998*, *2005*; *Fanselow, 1994*), such as freezing. Therefore, we initially asked whether freezing could be triggered by optogenetic stimulation of SF1[+] neurons. We found that when a 10-s blue light stimulation (20 Hz, 20 ms pulse width) was administered in the animals' home cage (*Figure 2A*), ChR2 virus-injected mice exhibited a short-latency (0.45 ± 0.09 s) freezing response (*Figure 2B–E,J*; *Video 1*). Control eYFP mice did not show any changes in behavior (*Figure 2E*).

In about 50% of animals, we observed that towards the end of the photostimulation period freezing behavior was followed by an activity burst, defined as a dramatic and sharply delimited

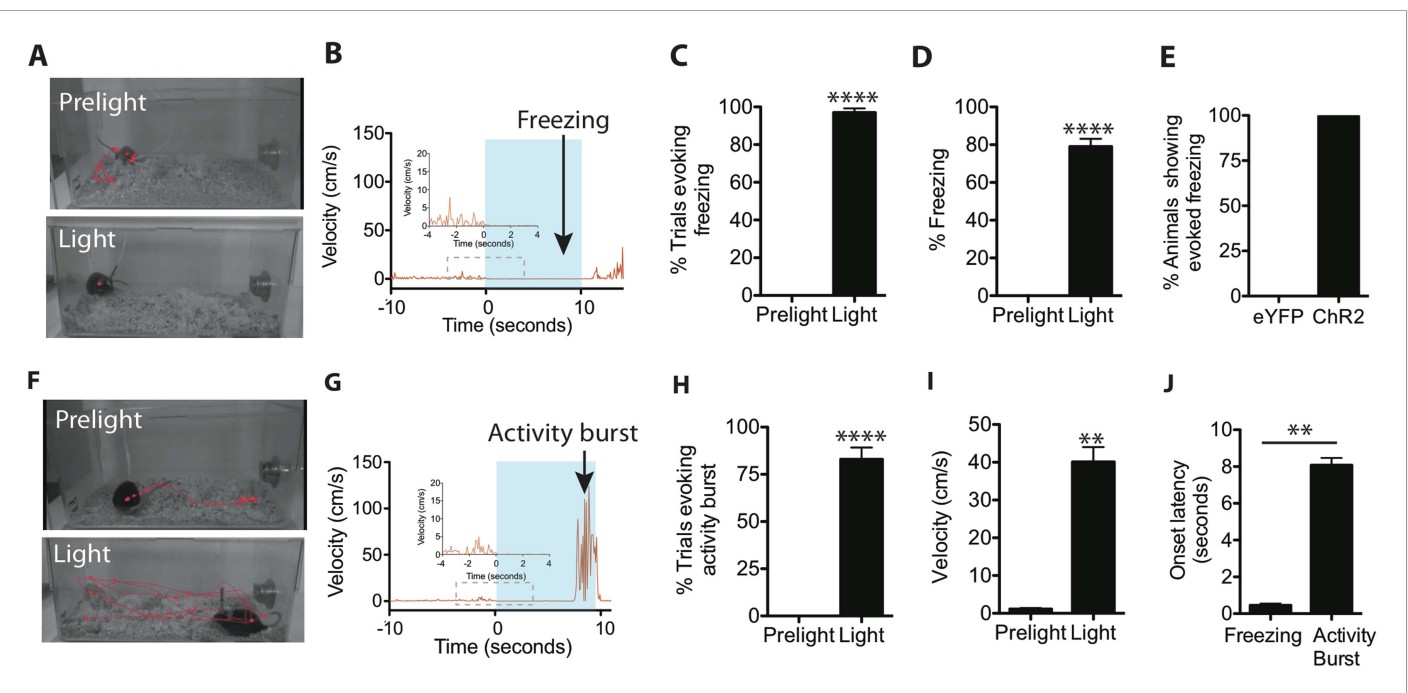

**Figure 2**. Optogenetic stimulation of SF1[+] neurons induces freezing and/or activity bursts. (**A**) Representative tracking traces (red) of SF1-ChR2 expressing mice before ('Prelight') or during ('Light') optogenetic stimulation. Red dot in lower image reflects immobility of animal. (**B**) Representative velocity trace displaying light-elicited freezing (arrow) in a ChR2 mouse. Blue shading represents period of photostimulation. Inset, expanded view of region in dashed box. (**C**) Percentage of photostimulation trials evoking freezing. (**D**) Percentage of time spent freezing during photostimulation averaged across trials. (**E**) Percentage of ChR2-expressing or control eYFP-expressing animals showing photostimulation-evoked freezing behavior. n = 17–18 animals for each group. (**F**) Representative tracking traces of a SF1-ChR2 expressing mouse ('Prelight') or during ('Light') optogenetic stimulation. Wider spacing between points in 'Light' indicates higher velocity. (**G**) Representative velocity trace displaying light-induced activity burst behavior in an SF1-ChR2 mouse. Note period of freezing prior to activity burst. Inset, expanded view of region in dashed box. (**H**) Percentage of stimulation trials evoking activity bursts following freezing. (**I**) Average velocity during activity burst period. (**J**) Average onset latency for freezing vs activity burst. n = 9 animals for each condition. Values are displayed as mean ± SEM. ****p < 0.0001; ***p < 0.001; **p < 0.01; *p < 0.05.

The following figure supplement is available for figure 2:

**Figure supplement 1**. ChR2-eYFP quantification in the VMHdm/c of freezing only and freezing + activity burst groups.

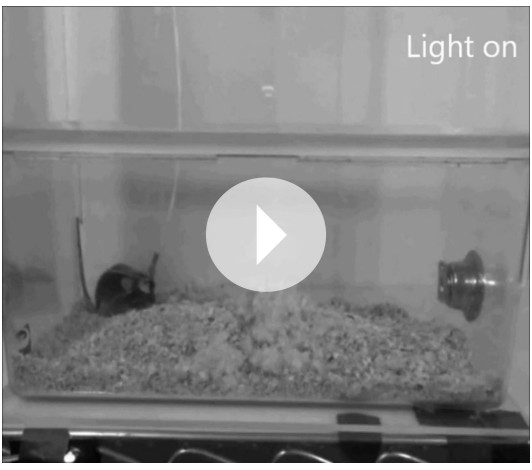

**Video 1.** ChR2-induced freezing.

episode of undirected, high velocity (>30 cm/s) movement often including vigorous jumping (*Figure 2F–J*; *Video 2*). This activity burst was similar to activity bursts observed in rodents exposed to a footshock (*Fanselow, 1982*; *Kiernan et al., 1995*) and is considered part of the repertoire of rodent defensive behaviors, presumably evoked in nature by a high intensity and/or proximate threat (*Anisman and Waller, 1973*; *Bolles and Riley, 1973*; *Fanselow and Lester, 1988*; *Fanselow, 1994*). Importantly, such activity bursts, when they occurred, were exhibited following a sustained (8.07 ± 0.39 s) period of freezing (*Figure 2G,J*). The activity burst terminated upon photostimulation offset (offset latency: 0.31 ± 0.04 s). Activity bursts were not observed in photostimulated control mice expressing eYFP in SF1+ neurons (data not shown), indicating that this behavior is not a consequence of heating the brain during the photostimulation trial.

We investigated the reason for this variability in activity burst behavior. Histological quantification of ChR2 expression revealed that mice showing activity burst responses exhibited a trend towards slightly higher levels of ChR2 expression in VMHdm/c, compared to animals showing freezing only (*Figure 2—figure supplement 1*). While this trend did not reach significance, it prompted us to investigate whether the level or intensity of SF1+ neuronal activation might influence the type of optogenetically evoked behavior.

To directly test whether activity bursts required a higher level of SF1+ neuron activation, we utilized SF1-Cre animals that were bilaterally injected with Cre-dependent AAV ChR2 and implanted with bilateral ferrule optic fibers, which enabled us to take advantage of the ability to independently stimulate one, the other, or both fibers within the same animal (*Figure 3A*). Assuming similar levels of ChR2 expression in each hemisphere, bilateral photostimulation should activate approximately twice as many SF1+ neurons as unilateral photostimulation (on either side) in the same animal. Indeed, episodes of freezing followed by activity bursts were only observed using bilateral stimulation (*Figure 3B*). Unilateral stimulation under these conditions evoked freezing but no ensuing activity bursts. These data suggest that activity burst behavior requires activation of more SF1+ neurons than does freezing.

To investigate further whether freezing required a lower level of SF1 neuron activation than did activity bursts, we systematically varied the intensity or frequency of optogenetic activation using bilateral photostimulation. The threshold light intensity required to evoke an activity burst was ~fivefold higher than that required for freezing (*Figure 3C–D*). Our previous studies of optogenetically evoked social behavior in VMHvl Esr1+ neurons indicated that more active neurons are recruited with increasing light intensity (*Lee et al., 2014*). Consistent with that conclusion, Fos was expressed in a significantly higher proportion of SF1+ neurons in animals that exhibited activity bursts (light intensity 5.25 mW/mm²), compared to those exhibiting only freezing (0.65 mW/mm²; *Figure 3E–F*). Activity bursts following freezing could also be elicited using a higher frequency of photostimulation at a fixed light intensity (*Figure 3G*), suggesting that increasing the

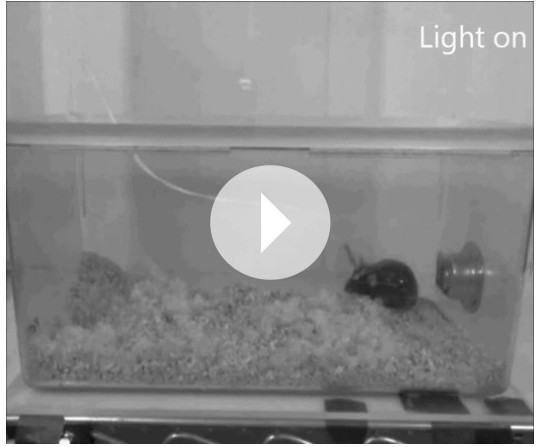

**Video 2.** ChR2-induced freezing and activity burst.

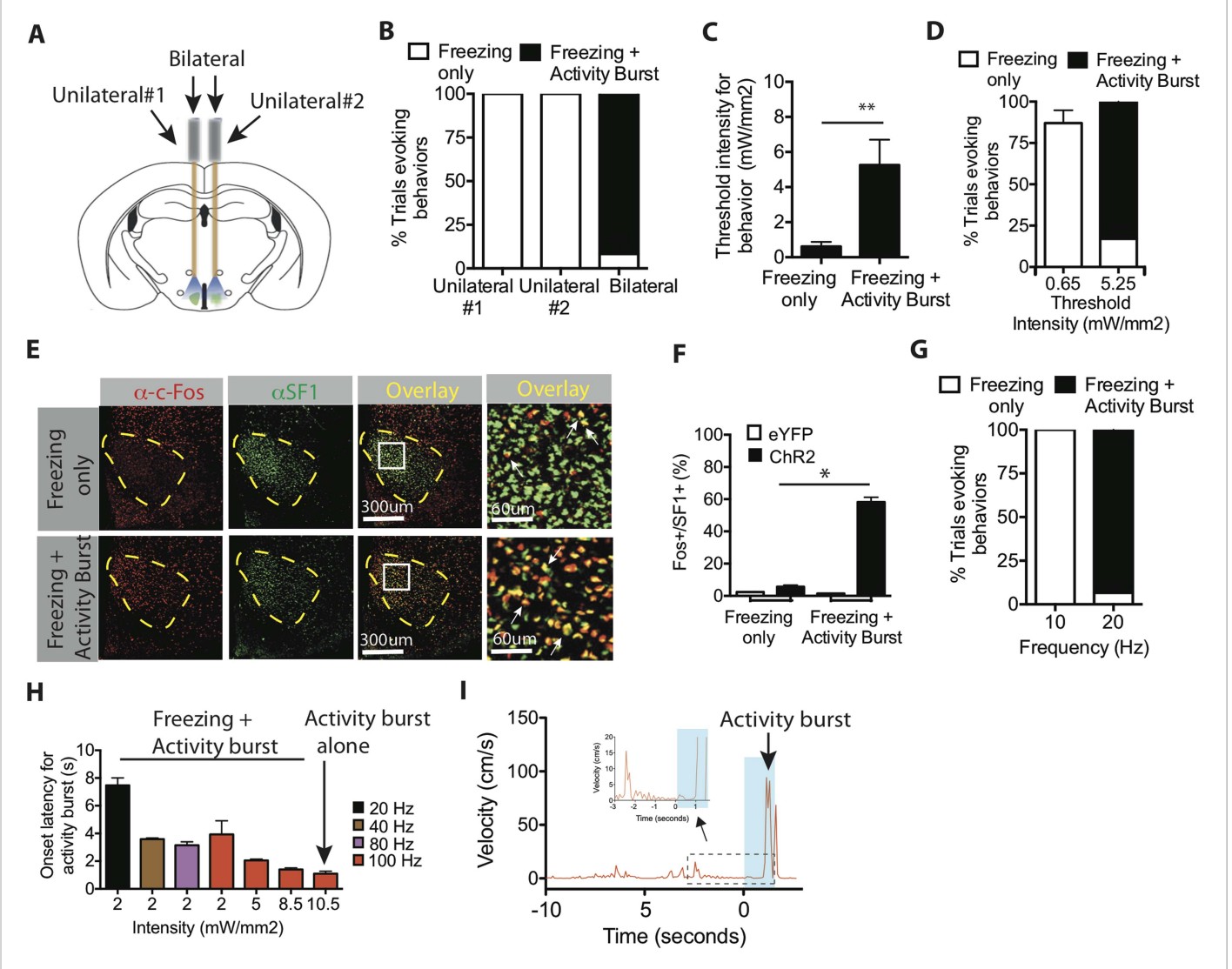

**Figure 3.** Optogenetic stimulation of SF1+ neurons induces freezing and/or activity bursts depending on strength and duration of photostimulation. (**A**) Schema illustrating unilateral vs bilateral optogenetic stimulation. Each mouse was implanted with bilateral optic fibers, and stimulation was delivered either to one side ('unilateral #1'), the contralateral side ('unilateral #2') or to both sides ('bilateral'). (**B**) Percent trials evoking freezing only (white) or freezing followed by an activity burst (black) in unilateral vs bilaterally stimulated ChR2 mice. **p < 0.01; Two-Way ANOVA, Bonferroni correction. n = 2 animals for each condition, each animal was stimulated either through one or the other of the two optic fibers ('Unilateral #1, Unilateral #2'), or through both ('Bilateral'). (**C**) Threshold stimulation intensity required to generate freezing alone or freezing followed by an activity burst, during the photostimulation period. n = 9 per condition. (**D**) Percentage of trials evoking freezing alone, or freezing followed by an activity burst, at respective stimulation intensities. **p < 0.0001; Two-Way ANOVA, Bonferroni correction. (**E**) Representative images of Fos+ (red) and SF1+ (green) neurons from animals exhibiting optogenetically induced freezing alone (upper), or freezing followed by an activity burst. Last column is higher magnification view of boxed area in adjacent 'Overlay' column. Arrow indicates cells double labeled for Fos+ (red) and SF1+ (green) (**F**) Percentage of SF1+ neurons that are Fos+ in eYFP (white) or ChR2 (black) mice following photostimulation trials eliciting freezing alone ('Freezing') or freezing followed by an activity burst ('Activity Burst'). n = 3–5 mice per condition. (**G**) Percentage of trials evoking freezing alone, or freezing followed by an activity burst, in response to different photostimulation frequencies. n = 5 animals per condition. (**H**) Onset latency for activity burst as a function of increasing light intensity (x-axis) and frequency (colored bars). Arrow indicates condition that elicited activity burst without prior freezing. (**I**) Representative velocity trace displaying light-induced activity burst without preceding freezing. Arrow indicates expanded trace from boxed region. n = 2 animals for each condition. Values are displayed as mean ± SEM. ****p < 0.0001; ***p < 0.001; **p < 0.01; *p < 0.05.

average spiking rate among these neurons can also shift the behavioral output from freezing towards the activity burst.

The observation that activity bursts are typically observed after several seconds of freezing (*Figure 3H–I*) raised the question of whether freezing behavior per se was a prerequisite for an activity burst. Therefore, we investigated whether certain stimulation conditions could elicit an activity burst without observable prior freezing. To do so, we used a more penetrant and highly expressing viral serotype, AAV5, for delivery of ChR2 to SF1+ neurons (*Aschauer et al., 2013*). Systematic manipulation of stimulation parameters yielded a high-intensity, high-frequency condition (10.5 mW/mm$^2$ and 100 Hz) that evoked a short-latency activity burst following stimulation offset, without prior freezing (*Figure 3H–I*, arrow). During the ramp-up to such a condition, the latency to the onset of the activity burst during freezing gradually decreased. Taken together, these data suggest that SF1+ neurons can trigger either freezing and/or activity burst behavior, depending on the intensity and duration of optogenetic activation.

## Activation of SF1+ neurons causes withdrawal

The undirected nature of the defensive responses evoked by photostimulation of SF1+ neurons in the animals' home cage left open the question of whether activation of these cells can promote avoidance or withdrawal. To investigate this question, we tested whether photostimulation of SF1+ neurons was sufficient to generate real-time place aversion (RTPA) (*Stamatakis and Stuber, 2012*; *Kim et al., 2013*). Mice expressing ChR2 in SF1+ neurons were randomly placed on one side of a contextually identical two-chamber place preference box (*Stamatakis and Stuber, 2012*) (*Figure 4A*). Photostimulation was delivered using a manual closed-loop protocol: the laser was switched on by the observer as soon as the mouse spontaneously entered the side opposite the one in which he had

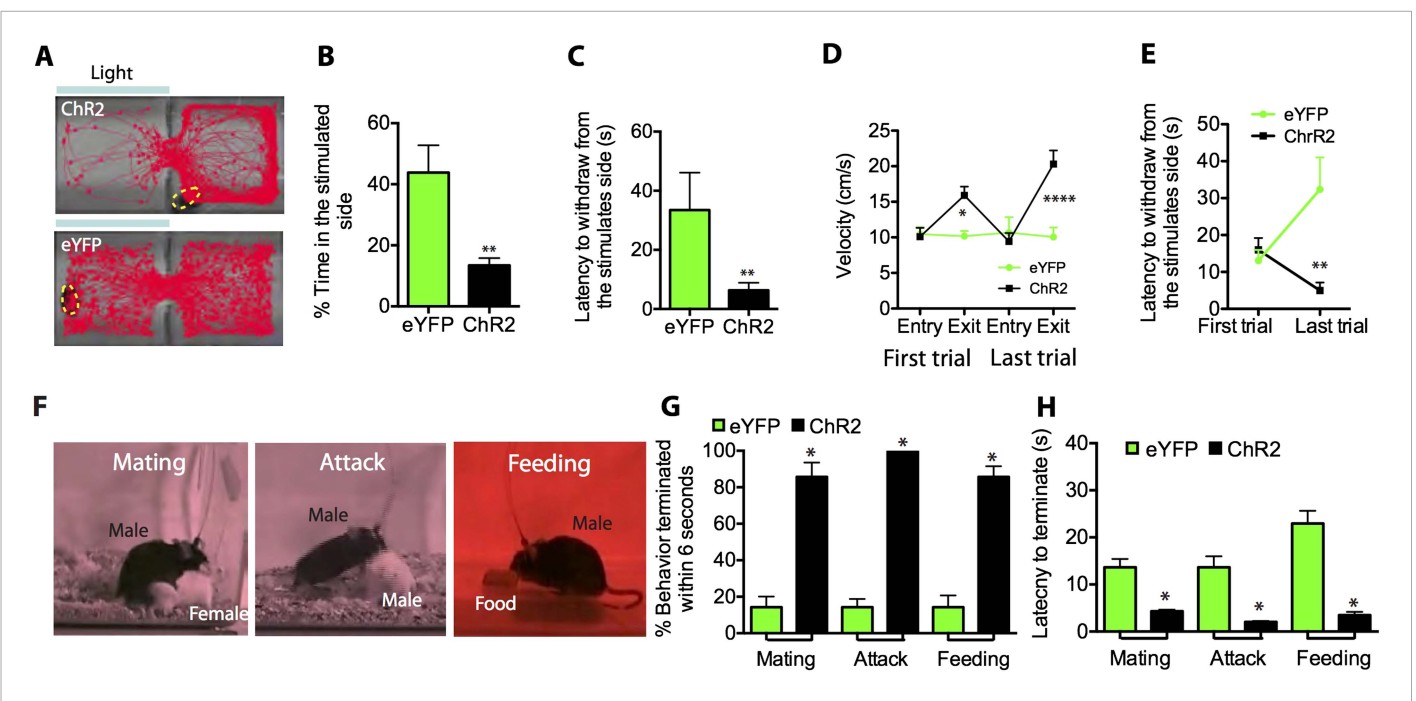

**Figure 4**. Optogenetic stimulation of SF1+ neurons induces aversion and interrupts ongoing consummatory behaviors. (**A**) Representative tracking traces of ChR2 mouse (top) and eYFP control mouse (bottom) in a real-time place avoidance assay (RTPA). Photostimulation (blue bar) was delivered in a manual closed-loop manner depending on the animal's behavior (see text). (**B**) Percentage of total time (20 min) spent in stimulated side during 20-min trial. (**C**) Average latency to withdraw from the stimulated side. (**D**) Average velocity to enter or exit the stimulated side for the first (left) vs last trial (right). (**E**) Latency to withdraw from the stimulated side for the first vs last trials. n = 6–7 animals for each condition in **D** and **E**. (**F**) Sample video still frames taken from consummatory behavioral assays. (**G**) Percentage of indicated behavior episodes terminated by light stimulation during the behavior within 6 s of photostimulation onset. (**H**) Latency to terminate respective consummatory behavior during photostimulation. n = 4–6 mice per condition. Values are displayed as mean ± SEM. ****p < 0.001; ***p < 0.001; **p < 0.01; *p < 0.05.

initially been placed; stimulation was continued until the animal moved to the non-stimulated side, at which point the laser was switched off. This stimulation regime was carried out over 20 min. Light pulses were delivered at low intensity ($0.01$ mW/mm$^2$), below the threshold required to elicit robust freezing or activity bursts.

During the stimulation period, ChR2-expressing mice spent significantly less time on the stimulated side of the chamber (*Figure 4B*), and withdrew from that side with shorter latency (*Figure 4C*) and higher velocity (*Figure 4D*) than did photostimulated eYFP-expressing control mice. Interestingly, the latency to withdraw from the stimulated side decreased significantly by the last trial for ChR2-injected mice compared to controls (*Figure 4E*; *Video 3, 4*), suggesting that sensitization of the withdrawal response occurred with repeated exposures to the stimulus. However, mice did not exhibit progressively longer latencies to re-enter the stimulated side across trials, suggesting that under these conditions no lasting association between photostimulation and the chamber was formed (but see below).

These results demonstrate that weak activation of SF1$^+$ neurons can produce withdrawal from a context distinct from the home cage. This suggests that SF1$^+$ neuron activation has a negative valence, in the behavioral sense of promoting avoidance rather than approach.

## SF1$^+$ photostimulation interrupts ongoing social and consummatory behaviors

In addition to simple avoidance, defensive states often actively inhibit positively valenced appetitive behaviors such as feeding or mating, which otherwise increase the animal's vulnerability to predation (*Petrovich et al., 2009*; *Sukikara et al., 2010*). Indeed, the sudden interruption of consummatory activity is often the most sensitive index of a perceived threat or anxiety state (*Estes and Skinner, 1941*). We thus investigated whether activation of SF1$^+$ neurons would terminate ongoing social and consummatory behaviors, including mating, aggression and feeding (*Figure 4F–H*). Indeed, all three behaviors were rapidly interrupted by modest photostimulation ($1.0$ mW/mm$^2$), in comparison to eYFP-expressing controls (*Figure 4G–H*). Although continued photostimulation under these conditions eventually resulted in freezing, appetitive behaviors could also be interrupted using shorter periods of photostimulation, during which freezing had not yet occurred (data not shown). These data indicate that SF1$^+$ neuronal activation not only promotes adaptive defensive behaviors, but also can abrogate consummatory and social behaviors.

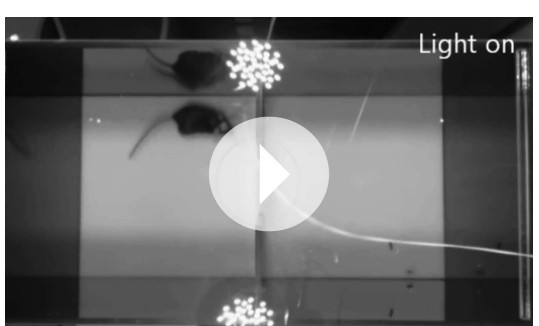

**Video 3.** eYFP control in RTPA.

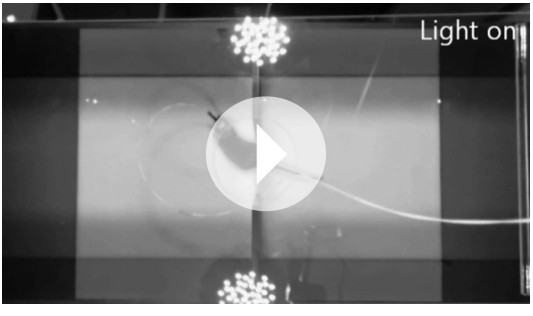

**Video 4.** ChR2–induced withdrawal in RTPA.

## Photo-elicited defensive responses are persistent

One hallmark of an emotional reaction is that it often persists beyond the stimulus that evoked it (*Blanchard and Blanchard, 1989a*, *1989b*; *Blanchard et al., 1990*; *Adamec et al., 2004*; *Anderson and Adolphs, 2014*). The preceding experiments indicate that SF1$^+$ neuron activation can evoke avoidance, freezing and activity bursts in a stimulus-bound and time-locked manner. We next investigated whether such activation could also produce persistent effects that endured beyond the photostimulation period.

Initial evidence of such persistence was observed during experiments in the home cage. We noticed that when optogenetic stimulation was terminated during an ongoing activity burst, the animal did not simply return to normal activity, but rather exhibited a period of freezing (*Figure 5A–C*; *Video 5*). Freezing typically occupied 40% of the time during a 10-s post-stimulation period (*Figure 5C*), and lasted from

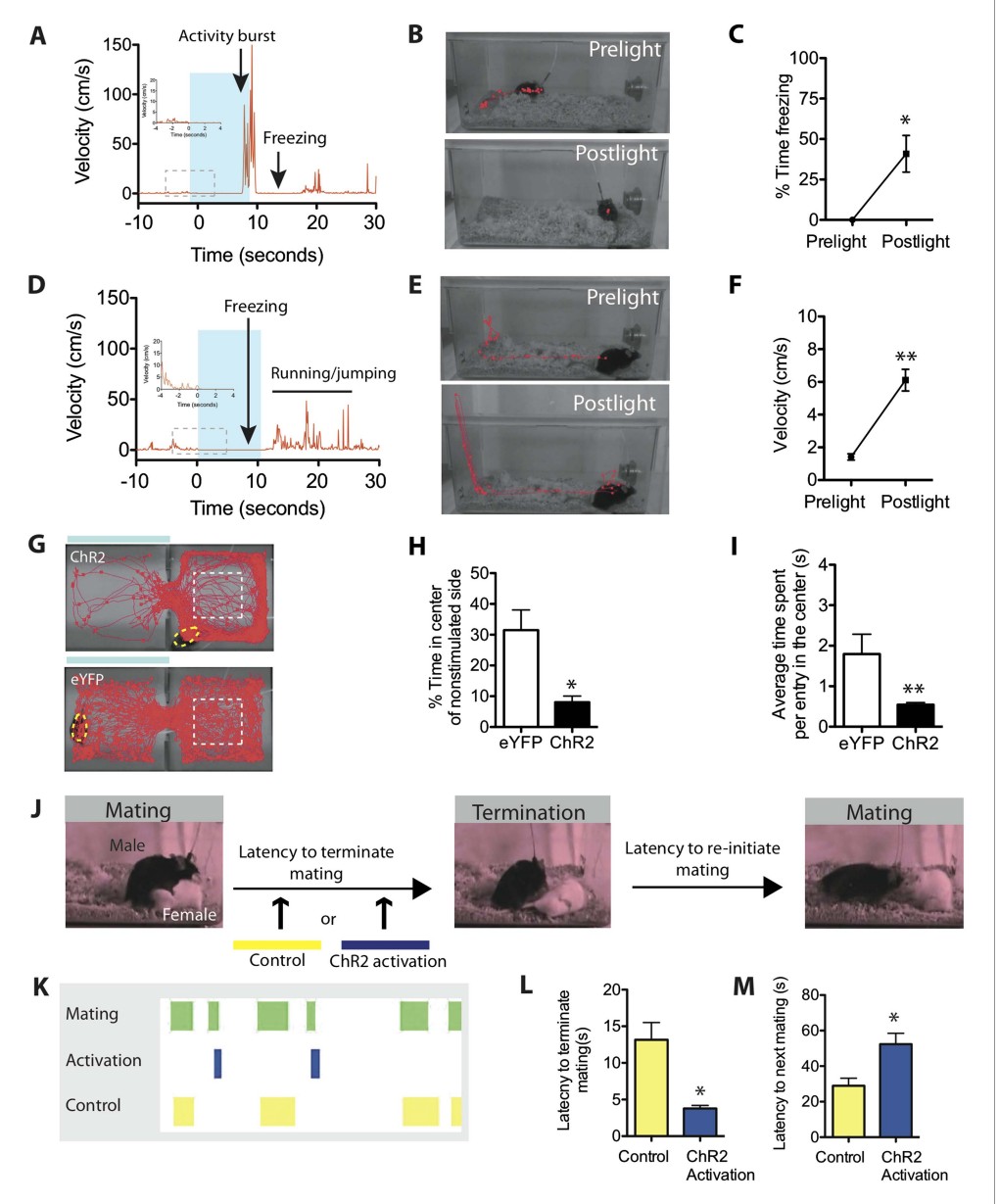

**Figure 5**. Stimulation of SF1+ neurons produces persistent defensive responses in multiple behavioral assays. (**A**) Representative velocity trace for ChR2 mouse displaying light-induced activity burst and post-light freezing behavior (arrow). Note that freezing was observed during the photostimulation period prior to the activity burst (inset, boxed region). See also *Figure 2G*. (**B**) Representative tracking traces (red) for pre-light vs post-light behavior in ChR2 mice. (**C**) Average percentage of time spent freezing during the 10-s pre-light and post-light bins averaged across trials and all mice. (**D**) Representative velocity trace for ChR2 mouse displaying freezing during photostimulation and elevated locomotion/jumping during the post-stimulation period. (**E**) Representative tracking traces for pre-light and post-light behavior in a ChR2 mouse. Vertical trace on left side of cage indicates jump ('Postlight'). (**F**) Average velocity during pre-light and post-light bins (10 s each) averaged across trials and mice. n = 9 animals for each condition. (**G**) Representative tracking traces of ChR2 mouse (top) and eYFP control mouse (bottom) in the RTPA assay. The white dashed box marks the center area of the non-stimulated side used to index thigmotaxic behavior. Modified from image in *Figure 4A*, to illustrate thigmotaxic behavior. (**H**) Percentage of total assay period (20 min) spent in the center of the non-stimulated side. (**I**) Average time spent in the center of the non-stimulated side, per individual entry. n = 6–7 animals for each condition. (**J**) Protocol to measure the resumption of mating behavior immediately following photostimulation of SF1+ neurons. (**K**) Representative raster plot illustrating mating episodes with ChR2 activation (blue) and control light (yellow) activation. The yellow wavelength

*Figure 5. continued on next page*

*Figure 5. Continued*

does not activate ChR2 and is used as an internal control. (**L**) Latency to terminate mating following photostimulation with blue vs yellow light. (**M**) Latency to re-initiate mating after mating termination following blue vs yellow light. n = 7 animals for each condition. Values are displayed as mean ± SEM. **p < 0.01; *p < 0.05.

The following figure supplements are available for figure 5:

**Figure supplement 1**. Persistent responses following light induced activity burst and freezing.

**Figure supplement 2**. Activation of SF1⁺ neurons produces an increase in neuroendocrine responding.

---

∼2–8 s depending on the animal (*Figure 5—figure supplement 1*). This observation suggests that photostimulation caused a period of residual defensive arousal, which gradually decayed over time.

Alternatively, if photostimulation was terminated while the animals were freezing, it was often followed by increased locomotion that was directed around the perimeter of the cage (average velocity ∼6 cm/s). This directed locomotion was qualitatively and quantitatively distinct from the undirected, high velocity (∼30 cm/s) activity burst behavior described earlier. Moreover, some animals exhibiting such post-stimulation increases in locomotor activity also attempted transiently to jump out of the cage (*Figure 5D–F*, *Video 6*; *Figure 5—figure supplement 1*). This increase in locomotor velocity decayed slowly, over a period of 40–60 s (*Figure 5—figure supplement 1*), while the period of jumping lasted only 15–20 s (*Figure 5D*, *Figure 5—figure supplement 1*). This latter observation suggests that post-stimulation induced jumping may require a higher level of residual defensive arousal than does increased locomotion.

We also investigated whether persistent effects of photostimulation could be observed in the real-time place avoidance assay. Notably, we observed that after ChR2 mice withdrew from the photostimulated side into the non-stimulated chamber, in contrast to control eYFP expressing animals, they avoided the center of the latter side, showing a dramatic increase in thigmotaxis (preferential occupation of the perimeter of an open space) (*Figure 5G–I*). Thigmotaxis is considered a key measure of anxiety (*Simon et al., 1994*). Thus, in both the home cage and in a different context, optogenetic stimulation of SF1⁺ neurons leads to persistent defensive or anxiety-like behaviors.

We also investigated whether transient activation of SF1⁺ neurons caused a persistent inhibitory influence on appetitive behavior, by testing whether it increased the latency to re-initiate mating, following its interruption (*Figure 5J*). We observed significantly longer latencies for ChR2 mice to re-initiate mating after SF1⁺ neurons were activated, compared to interleaved internal controls in which

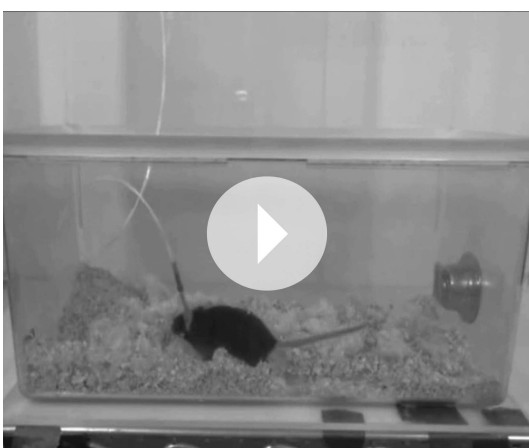

**Video 5.** ChR2-induced activity burst ands post-stimulation freezing.

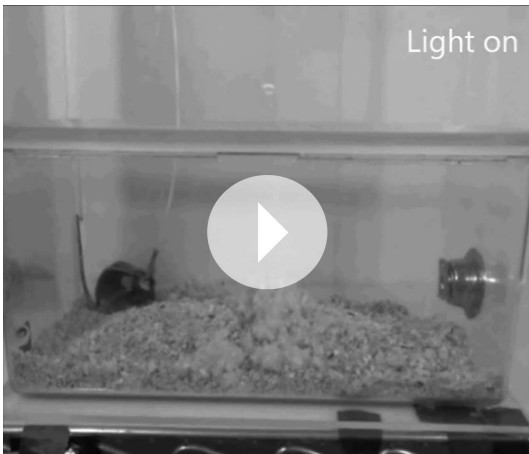

**Video 6.** ChR2-induced freezing and post-stimulation elevated locomotion and jumping.

yellow rather than blue light was delivered (*Figure 5K–M*). Thus, activation of SF1$^+$ neurons causes persistent defensive responses in a variety of behavioral assays, suggesting that it engenders an associated internal defensive state. Consistent with this idea, optogenetic stimulation of these neurons produced nearly a twofold elevation in serum corticosterone (*Figure 5—figure supplement 2*).

## SF1$^+$ neuron activation can condition learned avoidance behavior

There has been conflicting evidence as to whether hypothalamic stimulation can support conditioning, considered by some to be a *sine qua non* property of an emotion state (*Masserman, 1941*; *Cohen et al., 1957*; *Roberts, 1958*; *Wada and Matsuda, 1970*; *Panksepp, 2011a*, *2011b*). Therefore, we asked whether activation of SF1$^+$ neurons could serve as a US for the formation of associative fear memories. To do this, we used a modified two-chamber, real-time conditioned place aversion assay (RTPA) (*Figure 4A*), which we refer to simply as conditioned place aversion (CPA). We introduced several modifications for the CPA assay. First, we distinguished the two chambers by lacing them with different odors, and providing distinct mesh flooring and different colored plastic wall inserts. Second, for training we used bilateral stimulation with high light intensity (5.25 mW/mm$^2$), which evoked freezing and/or activity bursts, rather than the low-level stimulation employed for the RTPA assay, which produced withdrawal, but no associative memory (*Figure 4*).

The experimental design is illustrated in *Figure 6A*. Before conditioning, we performed a 5 min pre-training test to determine each animal's initial chamber preference. Photostimulation during training was then carried out on each animal's initially preferred side (IPS), to determine whether training would overcome this initial preference. Training was performed over a 20-min period, using the manual closed-loop procedure described for the RTPA assays (see above). It consisted of a series of photostimulation trials lasting 10 s, or until the animal withdrew from the stimulation chamber, whichever occurred first. Trials were repeated at 10-s intervals. If the animal withdrew from the training chamber during a photostimulation trial, the next trial was administered after the animal spontaneously re-entered the stimulation chamber. Over the 20-min training period, animals that initially exhibited freezing or activity bursts in the photostimulation chamber during the first few training trials eventually responded to photostimulation during later trials by rapid withdrawal from the IPS/training chamber.

Following the 20-min training period, animals were observed for an additional 10-min period and the percentage of time they spent in each of the two chambers during this period was measured. For convenience, we refer to this post-stimulation test period operationally as a 'short term memory' (STM) test. Following this STM test, animals were returned to their home cage for 24 hr and then tested for their chamber preference once again; we refer to this latter test as the 'long term memory' (LTM) test.

As expected, ChR2-expressing mice exhibited a dramatic avoidance of the photostimulation chamber during the training period, as compared to eYFP controls (*Figure 6B–C*). Importantly, robust avoidance of the stimulated (initially preferred) chamber persisted during the 10-min STM test (*Figure 6B–C*). This reversal of preference was reflected in a large negative value of the preference score, calculated as the percentage of time spent in the initially preferred chamber minus the percentage of time spent in the initially non-preferred chamber (*Figure 6D*) during the STM test (or other phase of the experiment). It was also reflected in a significantly more negative (in comparison to eYFP controls) difference score, calculated as the percent of time spent in the IPS during the STM test minus the percentage of time spent in the IPS before training (*Figure 6F*). (For both metrics, a negative value is indicative of avoidance of the initially preferred chamber). Other features of behavior, such as average velocity, were unaffected by conditioning (*Figure 6G*).

Re-testing the animals 24 hr later (LTM test) revealed a modest but statistically significant retention of conditioning in ChR2-expressing animals, compared to controls (*Figure 6B–C*) as indicated by both the preference score and the difference score (*Figure 6D–F*). Importantly, a comparison between pre-training and LTM preference scores (*Figure 6E*, LTM) revealed that the long-term reduction in preference for the IPS was robust (95% confidence interval: −72.30 to −18.14; Effect size: 1.2173; Power: 0.927). In contrast, eYFP control mice showed no significant change in preference score between the pre-training phase and the LTM test (*Figure 6E*). The retention of this conditioned response was confirmed by the significantly more negative LTM difference score, in

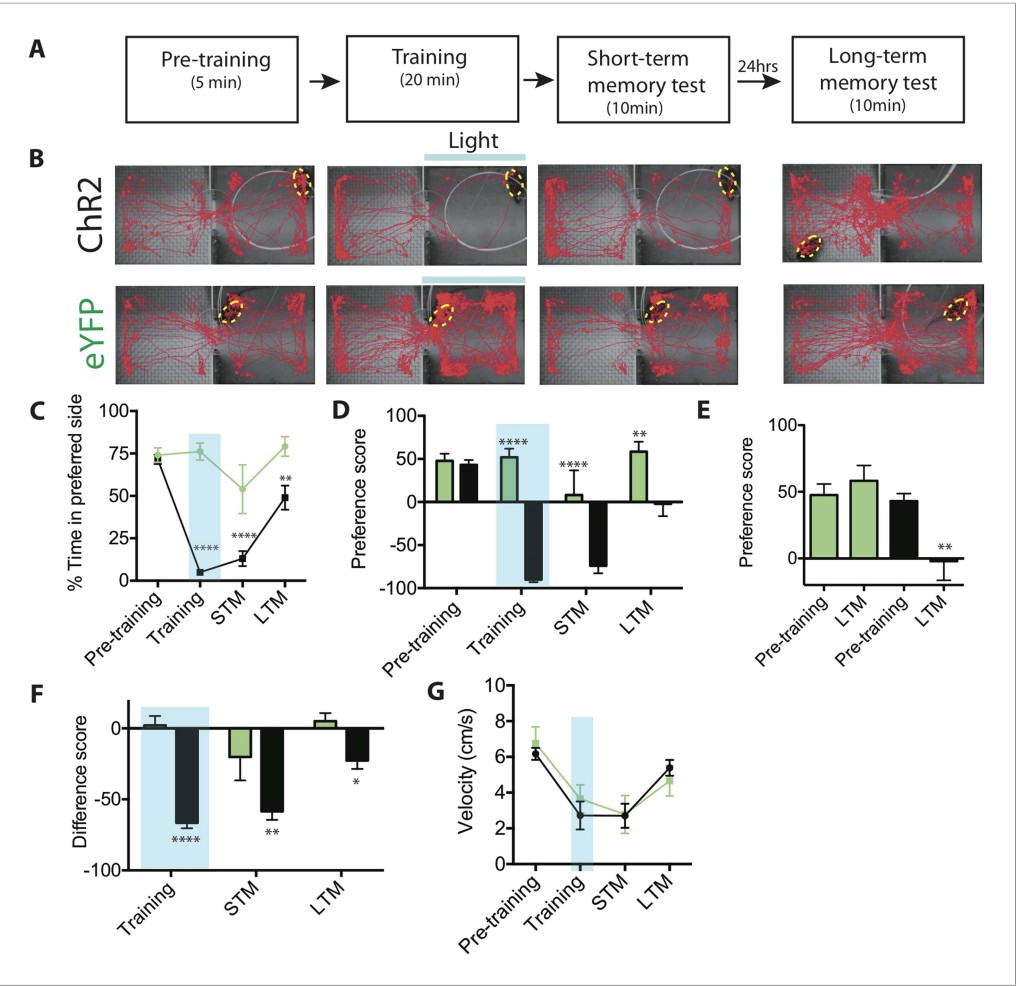

**Figure 6**. Activation of SF1+ neurons produces conditioned place avoidance learning and memory. (**A**) Protocol for conditioned place avoidance assay. Pre-training phase used to determine each animal's initially preferred side. (**B**) Representative tracking traces during pre-training, training, short-term memory (STM) and long-term memory (LTM) test phases as indicated in overlying schematic. Blue bar represents light delivery on the initially preferred side exhibited by each individual animal. (**C**) Percentage of total time for each phase (see panel **A**) spent in initially preferred side during pre-training, training (blue shading), STM test, and LTM test. Animals spontaneously spent ~75% of their time in one of the two chambers (defined as the initially preferred side), during pre-training period. (**D**) Preference score (percent total time spent in the initially preferred minus the initially non-preferred side), for each experimental phase. Negative value indicates that the animal spends more time in the initially non-preferred side, than in the initially preferred side. Preference score measures distribution of animals between the two chambers during a given testing phase. (**E**) Preference scores from (**D**) replotted for comparison of pre-training vs LTM scores for ChR2-expressing (black bars) vs eYFP control mice (green bars). (**F**) Difference scores (percent time spent in the initially preferred side during each respective phase minus the percent time spent in the same side during the pre-training phase) for training, STM and LTM tests. Difference score measures change in time spent in initially preferred side during pre-training phase vs a given testing phase. Difference score during LTM test is ~38% of that measured during STM test, indicating some retention of avoidance conditioning. (**G**) Mean velocity during each experimental phase revealed no differences. n = 7–10 animals for each condition. All values are displayed as mean ± SEM. ****p < 0.001; ***p < 0.001; **p < 0.01; *p < 0.05.

comparison to eYFP controls (*Figure 6F*). Based on a comparison of the difference scores between the STM and LTM tests (*Figure 6F*), ChR2 mice showed ~38% retention of their aversion memory. Thus, activation of SF1+ neurons can serve as a US for the formation of a conditioned place avoidance memory.

## Ablation of VMHdm/c SF1+ neurons attenuates both innate and conditioned defensive behaviors

The foregoing gain-of-function experiments raised the question of the context(s) in which the function of SF1+ neurons is normally required, and the precise role they play in such contexts. Previous studies have shown that the VMHdm/c is activated by predator cues (*Dielenberg et al., 2001*; *Martinez et al., 2008*; *Silva et al., 2013*; *Stowers et al., 2013*), and that SF1+ neurons are necessary for predator defensive responses, but not for other types of threat responses (*Silva et al., 2013*). These data have led to the view that VMHdm/c primarily mediates innate defensive responses to predators (*Gross and Canteras, 2012*; *LeDoux, 2012*). Alternatively, SF1+ neurons may control a defensive state that is employed in a broader variety of contexts.

To address this issue, we performed loss-of-function experiments to determine whether SF1+ neurons are required for a diverse set of threat-evoked defensive behaviors. Using a caspase-mediated cell ablation method (*Yang et al., 2013*), SF1+ neurons in the VMHdm/c were selectively killed. Animals were bilaterally injected with a Cre-dependent AAV encoding activated caspase3 (*Figure 7A*). We observed more than 90% elimination of SF1+ neurons in the VMH as compared to SF1-Cre negative littermates under similar conditions (*Figure 7B,D*). Adjacent cells in the VMHvl and ARH neurons were unaffected by this ablation, as shown by double labeling with the VMHvl and ARH-specific marker Estrogen receptor 1a (Esr1a) (*Lee et al., 2014*), confirming the restriction of caspase-mediated viral selective ablation to the SF1+ population in VMHdm/c (*Figure 7C,E,F*).

Following successful ablation, we tested mice across a variety of behavioral paradigms involving defensive behavior. Initially, we tested whether this manipulation could replicate the effect of pharmacogenetic silencing of SF1+ neurons to reduce predator responses (*Silva et al., 2013*), using a live rat as a stimulus (*Yang et al., 2004*; *Blanchard et al., 2005*). We found that males with ablated SF1+ neurons showed a striking deficit in predator avoidance behavior compared to controls, as demonstrated by a significant increase in the time spent in the zone closest to the rat (*Figure 7G,H*, Z1). In fact, some experimental animals even appeared to actively investigate the rat, hanging onto its mesh enclosure and attempting to poke their nose through the holes (*Video 7, 8*).

Next, we examined the behavioral effects of SF1+ neuronal ablation on auditory cued fear conditioning (*Figure 7J–L*) (*Wehner and Radcliffe, 2004*). Males with ablated SF1+ neurons were fear conditioned using five tone-conditional stimulus (CS) presentations co-terminating with a footshock-US. These mice showed a significant reduction in the magnitude of their activity bursts during the footshock, for some (but not all) of the training trials (*Figure 7K*). In addition, they exhibited a significant retardation in their ability to acquire conditional freezing to the tone-CS during training, as measured during the 30-s tone CS presentation prior to delivery of the footshock (*Figure 7L*). However, tone CS-induced freezing eventually reached the same asymptotic level as that observed in controls.

Lastly, we tested whether SF1+ neurons play an essential role in anxiety (*Gordon and Hen, 2004*; *Davis et al., 2010*). To do this, we compared the behavior of males with ablated SF1+ neurons vs controls in three different anxiety assays: the elevated plus maze, novel object test, and the light–dark box (*Gordon and Hen, 2004*) (*Figure 8A–C*). To increase baseline anxiety and avoid 'floor effects', animals tested in this assay were exposed to inescapable footshock 3 days prior to testing (*Maier and Watkins, 2005*). Ablation of SF1+ neurons significantly reduced measures of anxiety in each of the three different assays (*Figure 8D–F*). Neither entries nor velocity was significantly affected (*Figure 8G–L*), suggesting that this effect was not caused by differences in locomotor activity. Thus, SF1+ neurons are required for either the induction or expression of anxiety (or both).

Finally, we investigated the requirement of SF1+ neurons for defensive responses triggered by a visual threat. When presented for the first time with an overhead looming shadow, mice exhibit rapid and robust escape or freezing responses (*Yilmaz and Meister, 2013*). Interestingly ablation of SF1+ neurons did not significantly diminish these behavioral responses to the shadow (*Figure 8—figure supplement 1*). Thus these neurons are dispensable for rapid responses to a visual threat.

## Discussion

Defensive behaviors are prototypic emotional behaviors, but the relationship between the circuits that control such behaviors, and those that encode associated emotion states such as 'fear', (*Adolphs, 2013*) remains an open question. Here we have used optogenetic manipulations to probe the

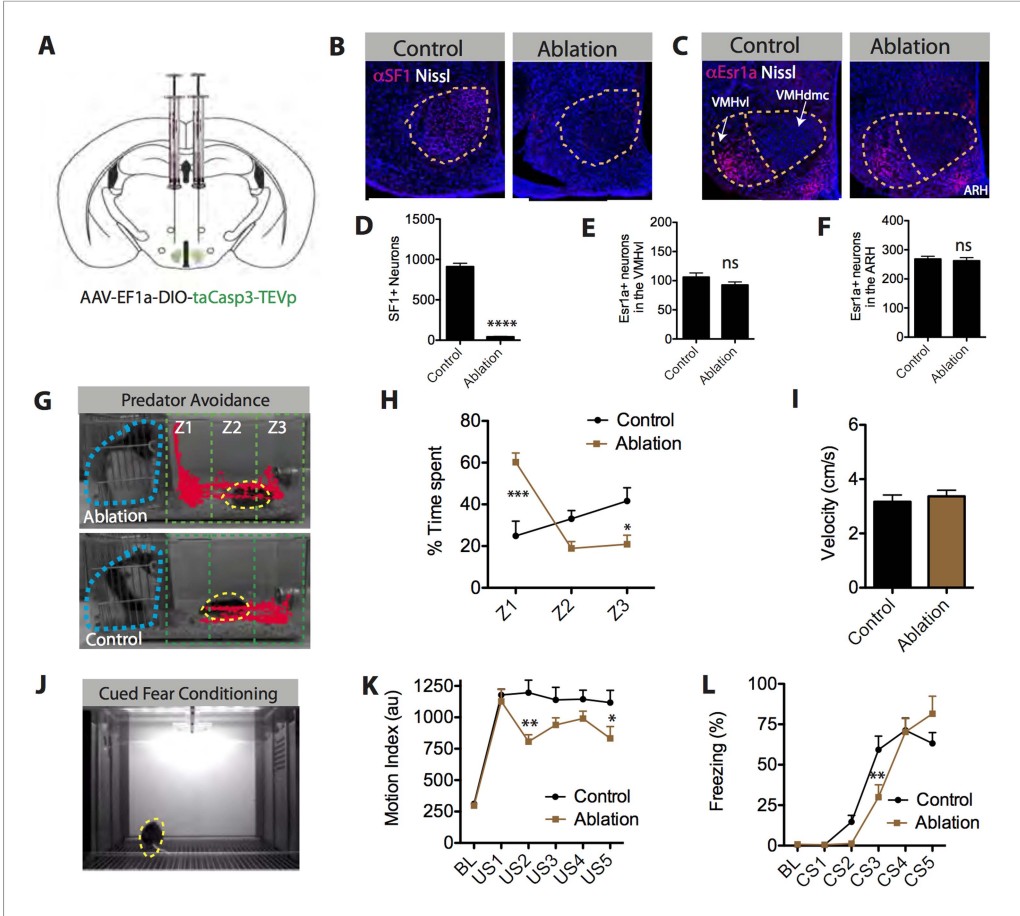

**Figure 7**. SF1⁺ neurons are necessary for predator aversion and conditional fear. (**A**) Schematic for bilateral injection of Cre-dependent apoptotic effector virus into the VMHdm/c of SF1-Cre mice. (**B**) Representative images of SF1⁺ neurons (red) in control and SF1⁺ ablated mice. (**C**) Representative images of Esr1a⁺ (VMHvl) neurons (red) in control and SF1⁺ ablated mice. (**D**) Number of total SF1⁺ neurons in VMHdm/c of control vs SF1⁺ ablated mice. (**E**) Number of total Esr1a⁺ neurons in VMHvl of control vs SF1⁺ ablated mice. (**F**) Number of total of Esr1a⁺ neurons in the arcuate nucleus (ARH) of control and SF1⁺ ablated mice. n = 4–6 animals, 3–4 sections per injection site. (**G**) Representative tracking traces of an SF1⁺ ablated mouse (top) and a control mouse (bottom) in a predator avoidance task. The rat predator (constrained within a mesh cage) is outlined in blue and the mouse in yellow. (**H**) Percentage of total test time (3 min) spent by mice in each zone, with Z1 representing the closest zone and Z3 the furthest. (**I**) Average velocity across entire predator avoidance test. n = 5–7 animals for each condition. (**J**) Still video frame from the cued fear conditioning assay. Mice were given 5 tone-footshock pairings. (**K**) Average motion index (au) units during the two-minute, pre-conditioning baseline ('BL') period and during the activity burst elicited by each footshock-US. (**L**) Average percent time spent freezing during baseline (BL period) and during each 30 s tone-CS presentation preceding delivery of footshock. n = 9–10 animals for each condition. Values are displayed as mean ± SEM. ***p < 0.001; **p < 0.01; *p < 0.05.

behavioral function of a genetically defined, highly specific and anatomically restricted hypothalamic neuronal subpopulation (*Dhillon et al., 2006*; *Silva et al., 2013*). Our results indicate that this population controls defensive behaviors in a manner which suggests that it implements an underlying causative emotion state (*Anderson and Adolphs, 2014*).

## SF1⁺ neurons control defensive behaviors in diverse contexts

The role of VMHdm/c neurons in the control of innate defensive responses has been investigated for over a quarter century (reviewed in [*Gross and Canteras, 2012*; *LeDoux, 2012*]). Early lesion or other inhibitory manipulations of VMHdm produced conflicting results regarding the valence of its role in

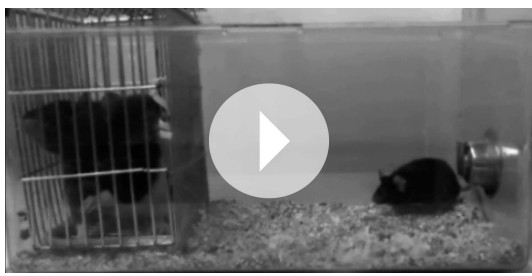

**Video 7.** Control ablated mouse in predator avoidance test.

defensive behaviors (*Turner et al., 1967*; *Grossman, 1970*, *1972*; *Weisman and Hamilton, 1972*; *Colpaert, 1975*). More recently, pharmacogenetic inhibition of SF1+ neurons mice was reported to cause a decrease in rat-evoked 'defensive behavior', a metric combining stretch postures and freezing or immobility (*Silva et al., 2013*). Here we used targeted genetic ablation (*Yang et al., 2013*) to address the necessity of these neurons for defensive behaviors. These loss-of-function experiments confirm a role for these neurons in predator avoidance, but also provide evidence of a broader role in defensive behaviors and associated emotion states.

In contrast to our findings, *Silva et al. (2013)* reported that chemogenetic inhibition of SF1+ neurons using DREADDs selectively impaired defensive responses to a predator (rat). The broader role of SF1+ neurons revealed by our loss-of-function experiments may reflect the fact that genetically targeted ablation (*Yang et al., 2013*) causes a more profound elimination of SF1+ neuronal function than does chemogenetic inhibition, whose extent and efficacy in vivo is difficult to assess. A caveat however, is that our genetically targeted ablation may have caused damage to neighboring populations of neurons, in a non cell-autonomous manner, and that such 'collateral damage', if it occurred, could contribute to some of the phenotypes reported here. However, our demonstration that the number of Esr1+ cells in the adjacent VMHvl and neighboring ARH is quantitatively unaffected following ablation suggests that this possibility is less likely. Furthermore, the observed reduction in predator defense behavior caused by ablation of SF1+ neurons (*Figure 7*) is similar to the effect of chemogenetic inhibition of these neurons reported earlier (although the behavioral metrics were different) (*Silva et al., 2013*). Together, these considerations suggest that the ablation of SF1+ neurons is responsible for the behavioral phenotypes we observed.

The role of VMHdm in defensive behavior has also been investigated previously using gain-of-function manipulations such as electrical or pharmacologic activation. In rats or non-human primates, such stimulation induced freezing and escape (*Lipp and Hunsperger, 1978*; *Lammers et al., 1988*; *Silveira and Graeff, 1992*; *Freitas et al., 2009*). However, the pharmacological methods (*Silveira and Graeff, 1992*; *Freitas et al., 2009*) lacked cellular specificity and high spatial resolution, while electrical stimulation (*Lipp and Hunsperger, 1978*; *Lammers et al., 1988*) could not exclude activation of fibers of passage. Moreover, loss- (*Silva et al., 2013*) and gain-of-function manipulations (*Lipp and Hunsperger, 1978*; *Lammers et al., 1988*; *Silveira and Graeff, 1992*; *Freitas et al., 2009*) were reported in different studies from different laboratories, using different species and different assays, making direct comparisons difficult.

Here we have performed both optogenetic activation, and targeted ablation, of a genetically defined subset of VMHdm/c neurons in mice, using a battery of behavioral assays including those where a predator was not present (e.g., anxiety assays). Our data argue that the function of SF1+ neurons is not restricted to predator defense per se, but rather that this population controls features of an internal defensive emotion state, which generalize across different contexts and different types of threats. Our results also argue against the more trivial interpretation that VMHdm/c serves exclusively as a permissive, sensory relay for predator-derived cues (*Papes et al., 2010*)—in essence, an 'internal nose'—a possibility that could not be excluded from earlier loss-of-function studies (*Silva et al., 2013*). However, our results do not exclude a role for SF1+ neurons in the transformation of sensory representations into an internal emotion or motivational state.

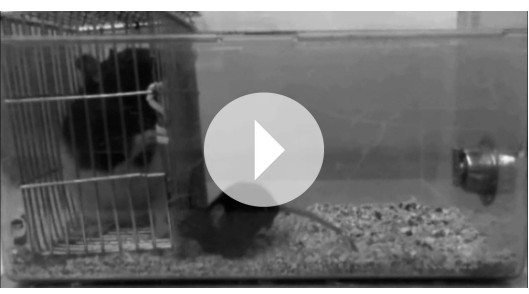

**Video 8.** SF1+ ablated mouse in predator avoidance test.

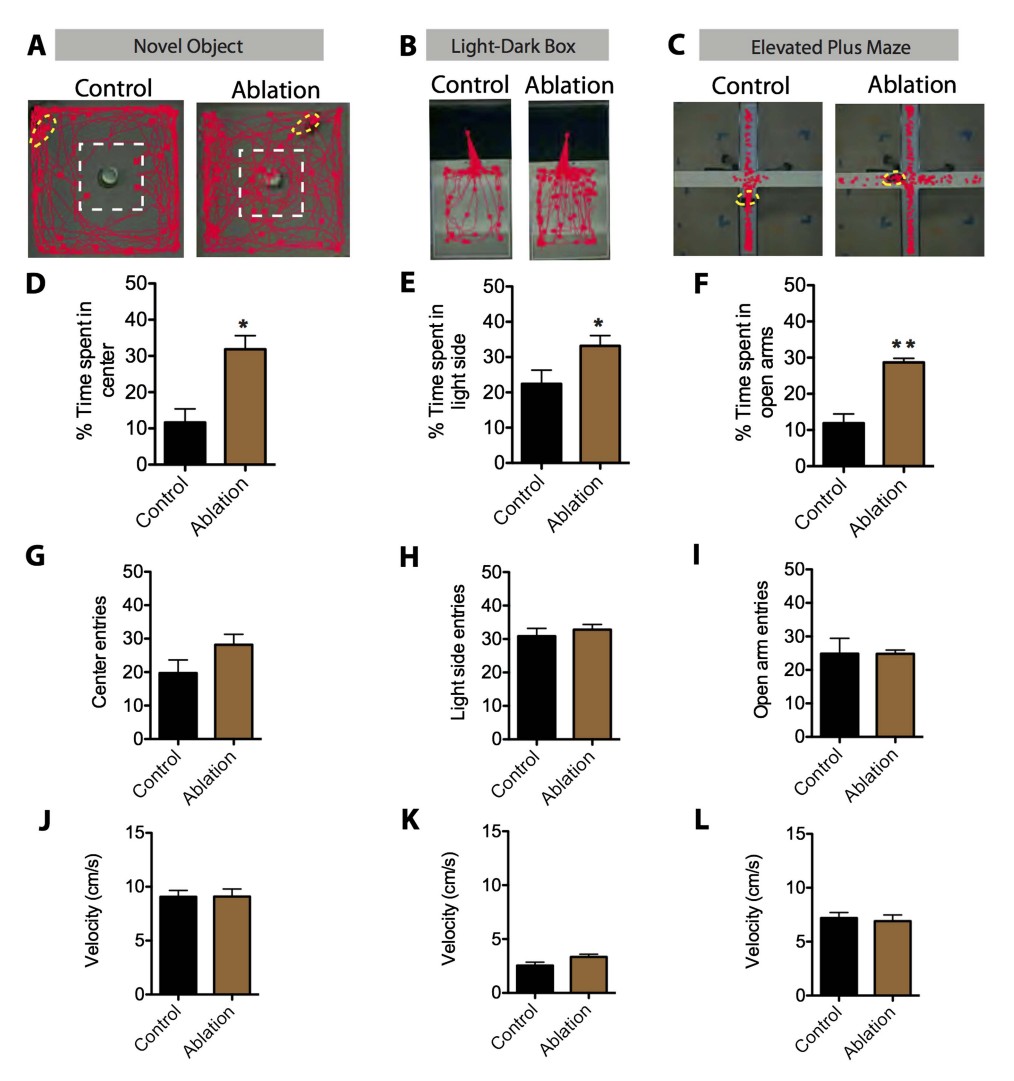

**Figure 8**. SF1+ neurons are necessary for anxiety. (**A**) Representative tracking traces in the novel object test for control (left) and ablated (right) mice. Mice are outlined in yellow. The dashed white box marks the center of the chamber. (**B**) Representative tracking traces in the light–dark box test. Note higher density of traces in the light side for the Ablation condition, in comparison to the control. (**C**) Representative tracking traces in the elevated plus maze. (**D**) Percentage of time spent in the center of the novel object test. (**E**) Percentage of time spent in the light side of the light–dark box. (**F**) Percentage of time spent in the open arms of elevated plus maze. n = 5–7 animals for each condition. (**G**) Center entries in the novel object test for control (black bars) and SF1-ablated (brown bars) mice. (**H**) Stimulated-side entries in the light–dark box assay. (**I**) Total open arm entries in the elevated plus maze. (**J**) Average velocity in the novel object test. (**K**) Average velocity in the light–dark box. (**L**) Average velocity in the elevated plus maze. n = 5–7 animals for each condition. Values are displayed as mean ± SEM. ***p < 0.001; **p < 0.01; *p < 0.05.

The following figure supplement is available for figure 8:

**Figure supplement 1**. SF1+ neurons are not necessary for defensive responses elicited by looming visual stimuli.

Interestingly, we observed that ablation of VMHdm/c SF1+ neurons did not impair innate freezing or flight evoked by an overhead expanding shadow (***Yilmaz and Meister, 2013***). This negative result could reflect redundancy in the circuits that mediate the shadow response. Alternatively, SF1+ neurons may play no role in this paradigm, suggesting that they either mediate defensive responses to

terrestrial but not to aerial threats, or that the visual response uses a specialized pathway, similar to the shadow-induced jump response in *Drosophila* (*Allen et al., 2006*). Whatever the explanation, these neurons are not essential for all forms of predator defense.

## SF1[+] neurons control different defensive behaviors in a time- and threshold-dependent manner

Changes in defensive behaviors during an encounter with a predator are often associated with a graded increase in the level of the underlying internal emotion or arousal state. These changes can be observed either as a quantitative increase in the amplitude or frequency of a given behavior (e.g., increased locomotor velocity), or as a qualitative shift in behavior (e.g., from freezing to flight) (*Fanselow and Lester, 1988*; *Blanchard et al., 1998*, *2001*, *2003a*; *Anderson and Adolphs, 2014*). Interestingly, the nature of the defensive behaviors evoked in this study depended on the intensity of the optogenetic stimulation: avoidance was evoked by low intensity stimulation, while freezing and the interruption of ongoing appetitive behavior required a higher intensity, and activity bursts yet more intense stimulation. By comparing unilateral vs bilateral stimulation of SF1[+] neurons in the same animal, we demonstrated directly that activation of a larger number of SF1[+] neurons was required to evoke an activity burst than to evoke freezing. Similarly, in the two-chamber assay, a higher level of photostimulation was required for associative memory formation, than simply to cause avoidance.

Similar threshold-dependent changes in behavior have been observed during optogenetic stimulation of medial amygdala and hypothalamic cell types that mediate social interactions (*Hong et al., 2014*; *Lee et al., 2014*), suggesting that it may be a general property of some behavior control circuits. Whether this scalable control is achieved through an increase in ensemble size and activity within a homogenous population of SF1[+] neurons, or reflects different subpopulations with different thresholds for activation (*Lee et al., 2014*), will be an interesting topic for future study. Whatever the answer, the observation that a common circuit node can control multiple defensive behaviors, according to its level of activity, argues against alternative views invoking parallel processing models, in which anatomically distinct pathways control different types of behavioral responses depending on cues or contexts (*Fanselow, 1994*; *Mobbs et al., 2007*).

Interestingly, we observed that photostimulation conditions that initially evoked freezing were often followed, after a delay of several seconds, by activity bursts during the stimulation period. This observation suggests that the brain may be able to integrate the cumulative effects of SF1[+] neuron activation over time, in a manner that changes the type of defensive behavioral output as different thresholds are reached. Such an integrative function is consistent with our observation that activation of these cells produces persistent behavioral effects, as persistent activity is a hallmark of neural integrators (*Major and Tank, 2004*; *Goldman et al., 2007*). Alternatively, the transition from freezing to activity burst in our experiments might reflect a time-dependent inactivation or habituation of freezing neurons during photostimulation, which in turn releases from inhibition a second population that controls the activity burst in an antagonistic manner. Whatever the explanation, the ability of SF1[+] neurons to integrate signals that change in their quality or intensity over time could allow an animal to express an appropriate behavioral response (freezing, escape) as a predator threat escalates, as encapsulated by 'Predatory Imminence' theories (*Fanselow and Lester, 1988*; *Blanchard and Blanchard, 1989b*; *Blanchard et al., 1998*, *2003a*; *McNaughton and Corr, 2004*). The neural mechanisms underlying such integration and persistent activity, and whether they are instantiated in VMHdm/c or in a downstream target, remain to be investigated.

## SF1[+] neurons, emotion and emotional learning

The prevailing, textbook view that the amygdala is the central orchestrator of emotion states (*Kandel et al., 2013*) is rooted deeply in its capacity to mediate forms of emotional learning, such as fear conditioning (*LeDoux, 1995*, *2000*; *Gallagher and Chiba, 1996*; *Maren and Fanselow, 1996*; *Fanselow and LeDoux, 1999*; *LeDoux, 2003*; *Phelps and LeDoux, 2005*; *Pessoa and Adolphs, 2010*). However this criterion is more difficult to apply to circuits that mediate unlearned (innate) defensive behavior. Indeed, the failure of hypothalamic electrical stimulation to condition learned defensive responses has been used to argue that the hypothalamus is not itself an emotion center (*Masserman, 1941*; *Wada and Matsuda, 1970*), despite some evidence to the contrary (*Cohen et al., 1957*; *Roberts, 1958*). Independent of learning, manipulations of VMHdm and other hypothalamic

nuclei in rodents have been interpreted as evidence that these structures control innate 'fear' (*Gross and Canteras, 2012*), a conclusion consistent with the observation that electrical stimulation of this region in humans evoked anxiety and panic attacks (*Wilent et al., 2010*, *2011*). However the attribution to animals of 'fear', a subjective human experience, has recently been questioned (*LeDoux, 2014*), on the grounds that it can only be assessed by verbal report in humans (*Adolphs, 2013*).

We have recently proposed objective criteria for identifying emotion states in animal models, based on general properties or features common to different emotions within a species, and to similar emotions across species (*Anderson and Adolphs, 2014*), and which are independent of anthropocentric attributions of human emotions such as 'fear'. These general properties include scalability, persistence, valence and generalization (*Russell, 2003*; *Posner et al., 2005*). The ability to mediate emotional learning is but one facet of these general properties, and not necessarily an essential one. If one accepts this view, then structures or neurons whose activation can evoke behaviors exhibiting these collective properties are good candidates for implementing emotion states.

The data presented here provide evidence that activation of SF1$^+$ neurons in VMHdm/c is able to evoke defensive behaviors exhibiting the aforementioned general features of an underlying causal emotion state. To our knowledge, this study is the first to provide evidence of an emotion state in an animal model, using the set of objective and general criteria described above (*Anderson and Adolphs, 2014*). In addition, we find that optogenetic activation of SF1$^+$ neurons in VMHdm/c can indeed serve as an unconditional stimulus (US) for associative learning, in a conditioned place avoidance assay. These data, together with earlier studies of conditioning in VMH (*Colpaert and Wiepkema, 1976*; *Santos et al., 2008*; *Santos and Brandão, 2011*) and associated hypothalamic nuclei (*Pavesi et al., 2011*), provide definitive evidence against the view that the hypothalamus is not an emotion center (*Masserman, 1941*; *Wada and Matsuda, 1970*). Yet this perspective is still common in textbook views of emotion ([*LeDoux and Damasio, 2013*], in [*Kandel et al., 2013*]), which place the amygdala as the central 'orchestrator' of emotion systems, and the hypothalamus as a motor effector or relay of amygdala output.

## The relationship between VMHdm/c and the amygdala in encoding emotion states

The data presented here demonstrate that direct, optogenetic activation of a specific hypothalamic cell population, in a manner that anatomically bypasses the amygdala, can evoke a persistent, scalable and generalizable emotion state. These observations argue that the prevailing, 'amygdalo-centric' view of emotion systems should be expanded to include specific hypothalamic structures such as VMHdm, and its associated circuitry (see below). While the VMHdm/c receives input from the anteriodorsal and posterioventral regions of the medial amygdala (MEAad and MeApv) (*Dong and Swanson, 2004*) and the basomedial amygdala (BMA) (*Petrovich et al., 2001*), recent data suggest that MeA functions primarily to encode sensory cues (*Bergan et al., 2014*). If so, then the transformation of such sensory input into an internal emotion state may, arguably, be carried out primarily at the level of VMHdm, or other interconnected hypothalamic nuclei (*Risold et al., 1994*), rather than in the amygdala itself. It should be noted the VMHdm also receives strong projections from the lateral parabrachial (PB) area, which transmits noxious stimuli (*Bester et al., 1997*); these projections may also provide sensory input to VMHdm important in the encoding of emotion states.

That said, we cannot formally exclude the possibility that the effects of optogenetically stimulating SF1$^+$ neurons are mediated by ascending (feedback) projections that activate the amygdala; in that case VMHdm/c would be 'upstream', rather than 'downstream', of the amygdala. However, high-resolution anatomical mapping of SF1$^+$ neurons indicates that recurrent projections to amygdala nuclei are relatively weak (*Figure 1—figure supplement 1* and http//:connectivity.brain-map.org, VMH, Nr5a1-Cre experiments 114290225 and 182337561, sections 64–82). This issue could be addressed, in principle, by combining bilateral activation of SF1$^+$ neurons with bilateral lesions of the amygdala. However such an experiment is challenging in mice because of the relatively small size of their brain, and the highly invasive nature of such an experiment. Thus, while descending input from the MEA is likely to contribute to VMHdm/c activation during defensive responses in an unmanipulated animal, our data show that one can experimentally bypass such amygdala input and evoke a persistent emotion state by direct activation of SF1$^+$ neurons.

## Are emotion states implemented within VMHdm/c itself, or by downstream structures?

VMHdm SF1[+] neurons lie within a densely interconnected network of hypothalamic and midbrain nuclei (*Canteras, 2002*; *Gross and Canteras, 2012*). Therefore the ability of SF1[+] neuronal activation to implement a persistent emotion state could be mediated by other nodes in this circuitry, rather than within VMHdm itself. VMHdm/c SF1[+] neurons send projections to the BNST, AHN, lateral hypothalamus (LHA), PMd, MeA and dorsal peri-aqueductal gray (dPAG), as well as to other structures (see *Figure 1—figure supplement 1*; *Video 9, 10*) (*Canteras et al., 1994*). Previous studies have shown that perturbations of some of these targets, including the dPAG or PMd, can influence some defensive behaviors (*Di Scala et al., 1987*; *Di Scala and Sandner, 1989*; *Canteras et al., 1997*; *Blanchard et al., 2003b*; *Bittencourt et al., 2005*; *Cezario et al., 2008*; *Pagani and Rosen, 2009*; *Sukikara et al., 2010*; *Pavesi et al., 2011*; *Santos and Brandão, 2011*; *Kincheski et al., 2012*). However, many of these earlier studies did not exclude a role for stimulation of fibers of passage, and lacked the cellular specificity and spatio-temporal resolution of the methods employed here. Furthermore, as in the case of VMHdm/c, the high degree of connectivity between these structures makes it difficult to ascribe specific functions to any individual node.

Among potential downstream targets that may mediate the effects of SF1[+] neuronal activation, the dPAG is a particularly noteworthy candidate. Activation of dPAG induces freezing and flight (*Brandao et al., 1982*, *1986*; *Vianna et al., 2001*; *Bittencourt et al., 2005*), supports conditioning (*Di Scala et al., 1987*; *Kincheski et al., 2012*), and induces fear sensation in humans (*Amano et al., 1982*). It may also play role in anxiety (*Gomes et al., 2014*) and interruption of other appetitive behaviors (*Sukikara et al., 2010*). In preliminary experiments we have observed that direct optogenetic activation of vGlut2[+] neurons in dPAG induces freezing and activity bursts (data not shown). Consistent with this, while this manuscript was in its final production stages, *Wang et al (2015)* reported that activation of SF1[+] projections to the dPAG evoked immobility. In contrast, activation of projections to the AHN evoked low-intensity escape responses. However, individual SF1[+] neurons collateralized to both the dPAG and the AHN (*Wang et al., 2015*), raising the question of how independent control of these different defensive behaviors can be achieved. Future work will clearly be required to resolve this issue.

## Conclusion

The results presented here characterize an important function for a specific, genetically defined hypothalamic cell type in promoting defensive behaviors, in a manner suggesting that these neurons induce or implement a persistent, scalable, generalizable, negatively valenced internal and causal emotion state. This state of apparent threat arousal or defensive motivation may share properties in common with the emotions that humans subjectively experience as 'fear' or 'anxiety'

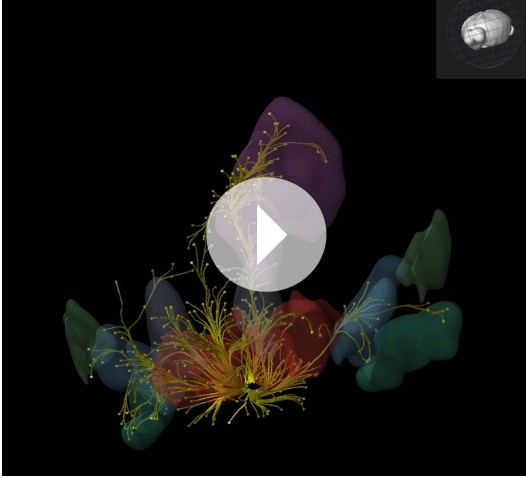

**Video 9.** SF1[+] projections.

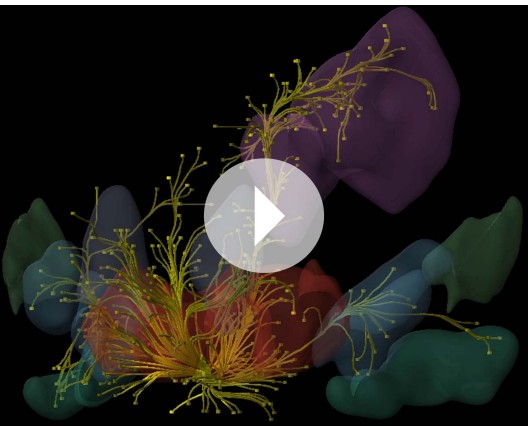

**Video 10.** SF1[+] projections II.

(*Phelps and LeDoux, 2005*; *LeDoux, 2012*; *Adolphs, 2013*; *Anderson and Adolphs, 2014*). While these results raise many important and unanswered mechanistic questions at the level of connectivity and neuronal activity dynamics, at the very least they should prompt a re-evaluation of the prevailing, 'amygdalo-centric' view of emotion control systems, by providing evidence that the hypothalamus is not simply a passive relay or effector of amygdala output, but can serve to implement a central emotion state itself.

## Materials and methods

### Ethics statement
These experiments were approved by the institutional animal care committee (IACUC) at the California Institute of Technology (protocol number 1602, 1600 and 1552).

### Mouse strains and husbandry
*SF1-Cre* mice were provided by Dr Brad Lowell (*Dhillon et al., 2006*) and backcrossed to C57Bl/6N wildtype mice (Charles River, Burlington, MA) at the Caltech animal facility. Heterozygous male mice or their littermates aged 12–20 weeks were used for behavioral studies. Heterozygous females aged 8 weeks were used for slice electrophysiology experiments. Mice were maintained on a reversed, 14-hr light cycle and all experiments were conducted during the dark cycle. Long-Evans rats aged 12–16 weeks were ordered from Charles River for use in the predator exposure experiment. All procedures described here adhere to the NIH guidelines for animal research.

### Viral vectors and stereotaxic surgery
AAV-EF1a-DIO-eYFP, AAV-EF1a-DIO-ChR2-eYFP, and AAV-EF1a-DIO-taCasp3-TEVp (*Yang et al., 2013*) were purchased from the University of North Carolina vector core facility. AAV-EF1a-DIO-mCherry (*Anthony et al., 2014*; *Lee et al., 2014*) was constructed in house (Dr Todd Anthony) and packaged by the University of Pennsylvania vector core facility. Mice were stereotaxically injected with viruses as previously described (*Cai et al., 2014*). Briefly, viruses were pressure injected (Mico4Controller, World Precision Instruments; Nanojector II, Drummond Scientific) unilaterally (*Figure 1O–U*, *Figure 4F–H* and *Figure 5J–M*) or bilaterally into the VMHdm/c using a pulled glass needle aimed at the VMHdm/c (ML ± 0.5, AP-4.65, DV-5.5) following a high resolution atlas (*Aravanis et al., 2007*). A total volume of 600 nl/site was injected at the rate of 100 µl/min. The needle was left in place for an additional 10 min to control for potential virus drag across the needle tract. A custom made bilateral ferrule fiber (200 µm in core diameter, Doric Lenses) or a unilateral cannula (24 gauge, Plastics One) was then placed 0.5 mm above the injection site. Fibers were cemented in place (Metabond). Following surgery, mice were allowed to recover on a heat pad and thereafter closely monitored for an additional 5 days during which they received medicated water (Septra and Motrin). Mice were single-housed for 4 weeks before commencing experiments to ensure surgical recovery and optimized Cre-mediated recombination.

### Optogenetic activation in the home cage
Animals were anaesthetized briefly using isoflurane to connect the fiberoptic cable to the unilateral cannula or bilateral ferrule. Mice were allowed to recover for 30 min in their home cage. They were then brought into an adjacent behavioral testing room for digital video capture of homecage behavior. The fiberoptic cable was then connected to a laser (473 nm for ChR2 stimulation and 593 nm for control stimulation, Shanghai laser) using a bilateral commutator (DoricLenses). A signal generator (World Precision Instruments) was used to control duration, frequency and pulse width of the light. A 20 Hz, 20 ms pulse width was used in all experiments except where mentioned otherwise. Laser intensity was calculated for a distance of 0.5 mm below the fiber tip.

Testing for freezing and activity burst behavior in the homecage was comprised of a period of baseline behavior recorded in the homecage followed by a series of optogenetic stimulations. Each stimulation was 10 s in duration, except in the case of stimulation-induced activity bursting, where the laser was turned off immediately upon the production of an activity burst, regardless of whether this period comprised less than 10 s. Each animal was given six stimulation 'trials' per optogenetic condition, with at least an average inter-stimulation-interval of 90 s. Behavior during stimulations was

averaged for data analysis. Freezing behavior was assessed by a complete lack of mobility except that required for respiration for 2 s or more using a custom designed behavioral scoring program in MATLAB (*Yang et al., 2013*). An activity burst was defined as a sharp, random movement with high locomotion (>20 cm/s of velocity, sustained for at least 1 s). Jumps were determined by assessing whether a mouse moved upwards with all four-legs off the ground. The total number of jumps occurring within the 20 s pre-, during, and post-light stimulation were calculated per mouse. Animals were rested for a week for subsequent tests.

### Optogenetic induction of Fos

Fos induction in response to optogenetic stimulation was assessed in ChR2 SF1-Cre mice receiving blue light stimulation (473 nm, 20 Hz 20 ms, 10 s on and 10 s off for 20 min) in their home cage. Following optogenetically-induced freezing and/or activity bursts, mice were kept in isolation and perfused 90 min later. Brains were extracted and harvested for subsequent sectioning and antibody staining.

### Social behavior testing

Interruption of aggression and mating behavior was tested using the resident intruder assay (*Hong et al., 2014*; *Lee et al., 2014*). Group housed, wildtype BALB/c males and females (Charles River) aged 12 weeks were used as intruders for aggression and mating testing, respectively. Females were selected for their receptivity beforehand to achieve robust baseline mating behavior. Resident males had at least 1 week of mating experience prior to surgery to increase their level of aggression (*Lee et al., 2014*). Resident males that failed to exhibit aggression or mating behavior were excluded from analyses.

To test for the ability to interrupt ongoing social behavior, mice were administered either blue (473 nm) or control yellow (593 nm) light (20 Hz, 20 ms) once a behavior was underway. Light was delivered until the behavioral episode was terminated. The test was continued until at least seven blue activation and yellow control trials were recorded. The order of ChR2 and control activations were counter-balanced across animals. The order of aggression and mating tests were counter-balanced across animals. Behavior in the resident intruder assay was recorded with a video camera mounted in front of the homecage and manually scored by an observer blind to experimental conditions. Scoring was performed using a custom designed behavioral scoring program in MATLAB (*Yang et al., 2013*).

### Feeding behavior

Following mating and aggression testing, mice were tested for interruption of feeding behavior as described previously (*Cai et al., 2014*). Mice were food-deprived for 24-hr, placed into a novel cage, and presented with a food pellet. Feeding behavior was interrupted using a stimulation protocol identical to that used for mating and aggression interruption (see above). Behavior was recorded using a video camera and subsequently scored by an observer blind to experimental conditions using a behavioral annotation software tool written in MatLab (*Yang et al., 2013*).

### Real-time place avoidance

Real-time place avoidance (RTPA) was performed as described previously (*Stamatakis and Stuber, 2012*). The apparatus (100 × 50 × 25 cm; black pexiglass wall; white pexiglass floor) was comprised of two identical sides that were connected by an opening (12.5 cm) in the center. Animals were placed pseudo-randomly in one side of the chamber (starting side was counterbalanced across mice) and blue light (20 Hz, 20 ms, 0.01 mW/mm$^2$) was delivered as soon as the mouse entered the alternate side of the apparatus by at least 50% of its body. Stimulation continued until the animal returned to the non-stimulated control side. The assay lasted a total of 20 min. Behavior during the session was recorded using a camera mounted above the apparatus and analyzed recording using Mediacruise recording software (Canopus). Total time spent in each chamber, chamber entries, and latency to depart the chamber following stimulation using Ethovision.

### Conditioned place avoidance

A custom-designed RTCPA apparatus was built for use in our conditioned place avoidance (CPA) assay. The apparatus measured 100 × 50 × 25 cm in dimensions. The two sides of the apparatus were

made contextually distinct. One chamber side was covered with black plastic and fine mesh flooring while the other side was left white and had coarse mesh flooring. The different sides were also distinguished by odor (2.5% of Acetophenone or ethyl acetate). CPA testing was carried out over 2 days. Day 1 involved a 5-min pre-training session to habituate animals to the apparatus and to determine each mouse's place preference. Animals that showed more than 90% of preference during pre-training were excluded from the analysis. Each animal was pseudo-randomly placed in one side (counterbalanced across mice). This was followed by a 20-min 'conditioning' session in which blue light stimulation (20 Hz 20 ms, 10 s on and 10 s off, 5.5 mW/mm$^2$) was administered in the preferred side until the mouse returned to the non-preferred side. Following conditioning, animals were allowed to move freely for another 10 min without light stimulation to determine their post-stimulation preference. After 24 hr, mice were returned to the apparatus to test for long-term aversion memories. Mice were placed in the stimulated side in order to access the aversion memory associated with the context. As in the RTPA task, time spent in each side during the CPA assay was assessed using Ethovision.

## Predator avoidance testing

Mice with ablated SF1+ neurons in VMHdm/c were used to test for intact predator avoidance (*Blanchard et al., 2005*). Predator rats weighting 300–500 gm were used to induce avoidance. A custom made testing apparatus measuring (36 × 18 × 40 cm) was designed to be attached to a mouse's homecage. The test rat was confined to a mesh enclosure (16 × 11 × 15 cm) and put on one side of the home cage. D-amphetamine (5.0 mg/kg, Sigma) was injected (i.p.) 20 min prior testing to trigger uniform movements in the rat stimulus. Rats were lowered into the mesh enclosure and mouse behavior was assessed across a 3-min time period. In order to assess how much time a mouse spent close or far from the rat predator, the home cage was divided into three equal 'zones', with Zone1 being closest to the rat and Zone 3 farthest. Time spent and frequency of entries into each zone was calculated using EthovisionXT software (Noldus).

## Auditory cued fear conditioning

SF1-ablated and control mice were placed in a conditioning chamber (Med Associates) and fear conditioned as previously described (*Haubensak et al., 2010*). After 2 min of habituation (baseline period, 'BL'), five training trials were delivered with an inter-trial interval of 1 min. Each trial consisted of a 85 dBA, 2k Hz tone for 30 s that co-terminated with a 2-s, 0.6 mA foot shock. Freezing and activity burst (measured by motion index) responses to the tone and shock, respectively, were analyzed using Video Freeze software (Med Associates).

## Anxiety tests

Open field, novel object, light–dark box and elevated plus maze tests (*Anthony et al., 2014*; *Cai et al., 2014*) were utilized to assess levels of anxiety in SF1-ablated mice. Mice were tested in the above-mentioned sequence of tests with at least a 4-hr rest period in between tests. The novel object test, which lasted 5 min, was done after the open field test. A stainless steel cup was placed at the center of the box. The center area for the novel object comprised 25% of total area. Time spent in the center as opposed to the borders of the apparatus was assessed. In the light–dark box, animals were initially placed on the light side of box and behavior was assessed across 10 min. In the elevated plus maze, animals were initially placed at the center of the maze and behavior was assessed across 10 min. Ethovision software was used to analyze time spent, entries, and velocities for each anxiety test.

## Looming visual stimuli test

SF1-ablated mice were tested for behavior in a looming visual stimulus test, as described elsewhere (*Yilmaz and Meister, 2013*). Wild type littermate sibling mice were used as controls. Animals were placed in an open-top pexiglass box (48 × 48 × 30 cm). A triangular shaped nest (20 × 12 cm) was placed in one corner. Recording using Nerovision software was done under illumination provided by Infrared LEDs (Marubeni). After 10 min of habituation, a looming stimulus was provided from above when an animal was in the center. The stimulus of 0.5-s duration was repeated 10 times with an inter-stimulus interval of 0.5 s. Mice were given a post-stimulation period of 10 min.

## Behavioral recording and analysis

Behaviors were recorded using Nerovision software control or Mediacruise recording software (Canopus). Annotation was carried out manually on a frame-by-frame basis by an experimenter blind to experimental conditions using a behavioral annotation software tool written in MatLab and/or using EthovisionXT.

## CORT measurements

Corticosterone measurement was done as described previously (*Anthony et al., 2014*). Mice were photostimulated with 473 nm light at 5.5 mW/mm$^2$, 20 Hz, 20 ms pulse with 10 s on and 10 s off for 30 min, immediately euthanized and decapitated for blood collection for corticosterone measurement using an immunoassay (Enzo Life Sciences).

## Immunohistology and cell counting

Sectioning and immunostaining were done as described previously (*Haubensak et al., 2010*). Mice were perfused using 4% PFA. Brains were cryoprotected (15% sucrose) and frozen at −80°C until sectioning. Brains were sectioned on a cryostat (Leica, Biosystems) at either 30 μm thick (for direct mounting onto slides) or 60 μm thick (for free-floating sections collected in a staining disc). The following antibodies were used: rabbit anti-SF1 antibody (1:500, TransGenic), rabbit anti-SF1 antibody (1:500, Upstate), goat anti-c-Fos (Santa Cruz, 1: 500), rabbit anti-Esr1a (1: 500, Santa Cruz), mouse anti PR (1:500, Thermoscientific), rabbit anti-GFP (1:500, Invitrogen). The following secondary antibodies were used—donkey anti-goat IgG Alexa 546 (1:500, Invitrogen), goat anti-rabbit IgG Alexa 488 (1:500, Invitrogen), goat anti-rabbit IgG Alexa 546 (1:500, Invitrogen), goat anti-rabbit IgG Alexa 633 (1:500, Invitrogen). NeuroTrace fluorescent Nissl stains (1:200, Invitrogen) or DAPI (1:200, Invitrogen) was used to counterstain sections and label cell bodies. At least three representative coronal sections spaced equally along the AP axis were used for quantifications.

## Statistics

Prism 5 (GraphPad) software was used for statistical analysis of behavioral and histological data. Data are presented as mean ± sem. The cutoff set for significance for all experiments was alpha <0.05. Data were tested for uniform distribution using three different normality tests (Kolmogorov–Smirnov test, D'Agostino and Pearson omnibus normality test and Shapiro–Wilk normality test). If data passed these normality tests, parametric tests were used. Otherwise, non-parametric tests were used. Unpaired t tests and Mann–Whitney tests were used for comparison between subjects, and paired t tests and Wilcoxon matched-pairs signed rank tests for within-subjects comparisons. For data involving two or more independent variables, two-way ANOVAs were used and Bonferroni posthoc tests, correcting for multiple comparisons, were used.

## Electrophysiological slice recordings and optogenetics in vitro

Brain slices were prepared from 3-month-old mice approximately 4 weeks after virus injection (*Haubensak et al., 2010*; *Cai et al., 2014*). Coronal brain sections of 250 μm thickness were cut with a Leica VT1000S vibratome, using ice-cold glycerol-based ACSF containing (in mM): 252 glycerol, 1.6 KCl, 1.2 NaH$_2$PO$_4$, 1.2 MgCl$_2$, 2.4 CaCl$_2$, 18 NaHCO$_3$, 11 Glucose, oxygenated in carbogen (95% O$_2$ balanced with CO$_2$) for at least 15 min before use. Brain slices were recovered for ~1 hr at 32°C and then kept at room temperature in regular ACSF containing (in mM): 126 NaCl, 1.6 KCl, 1.2 NaH$_2$PO$_4$, 1.2 MgCl$_2$, 2.4 CaCl$_2$, 18 NaHCO$_3$, 11 Glucose, oxygenated with carbogen. The fluorescence of the SF1$^+$ neurons was detected by a fluorescence video microscopy camera (Olympus BX51). Whole-cell voltage or current clamp recordings were performed with a MultiClamp 700B amplifier and Digidata 1440A (Molecular Devices). The patch pipette with a resistance of 5–8 MΩ was filled with an intracellular solution containing (in mM): 135 potassium gluconate, 5 EGTA, 0.5 CaCl$_2$, 2 MgCl$_2$, 10 HEPES, 2 MgATP and 0.1 GTP, pH 7.2, 290–300 mOsm. Data were sampled at 10 kHz, filtered at 3 kHz and analyzed with pCLAMP10 software.

Optogenetic photostimulation was delivered by a 473 nm laser (Shanghai Dream Laser, 473 nm) controlled by an Accupulser Signal Generator (World Precision Instruments). The estimated power at the specimen was set to 1 mW/mm$^2$, as measured with a photodiode (Thorlabs).

## In vivo recordings

In vivo electrophysiological recordings were performed using custom-built electrode bundles or optrodes, as published before (*Lin et al., 2012*; *Lee et al., 2014*). The electrode bundle was affixed to a movable microdrive stage that could be lowered in steps of 55 μm when required. The electrode bundle was implanted to the stereotaxic coordinates that correspond to the dorsal extent of the VMHdm and lowered at least one step for every recording session.

For single unit recordings during optogenetic photostimulation, we integrated a 62.5 μm core optical fiber into the 16-microwire electrode bundle (*Lee et al., 2014*). Data was collected from neurons in the SF1-Cre mouse line with the expression of ChR2 using AAV2.EF1α.FLEX.ChR2-eYFP, as identical to that used in the behavioral experiments. Photostimulation parameters for a given optrode were calibrated prior to implantation so that the transmitted light would irradiate the brain tissue at 1.0–1.5 mW/mm$^2$, measured under constant illumination. Hardware and software provisions for eliminating photoelectric artifact were used (*Kvitsiani et al., 2013*). All spikes recorded at a single microwire electrode crossing a threshold two standard-deviations over baseline (spike wavelength around 1 ms and interspike interval greater than 2 ms) were sorted into clusters using PC analysis and were considered to represent individual units. Units were recorded under the same photostimulation parameters as those used in the behavioral experiments that is, at 20 Hz with 20 ms pulse-width. Neural activity was recorded over a baseline period of 40 s, followed by a photo-stimulation period of 40 s.

## Acknowledgements

The authors thank Monica McCardle, Heeju Kim, Jung-Sook Kim and Xiaolin Da for technical assistance; Celine Chiu and Gina Mancuso for administrative support; and Robert Robertson, Brian Duistermars and Allan Wong for assistance with Matlab programming. We thank all members of Anderson lab for sharing regents and constructive discussion, Dr Todd Anthony for providing viral reagents and Dr Brad Lowell for SF1-Cre mice. We thank Ann Kennedy for assistance with data analyses, and comments on manuscript. A Steele and P Paterson laboratories for discussion, sharing reagents and providing technical help. Dr Baer, Dr Lencioni, Jennifer Constanza, Jeffrey Cochrane, Ruben Bayon, Ana Colon and Sarah Fitzgerald for ensuring animal welfare, care and husbandry. We would also like to thank R Mooney, R Adolphs, MS Fanselow, R Malenka and A Choe for constructive comments on the manuscript. Lastly, we would like to dedicate this manuscript to Dr Robert J Blanchard (d. 2013), who inspired this project and many others.

## Additional information

### Funding

| Funder | Grant reference | Author |
| --- | --- | --- |
| National Institutes of Health (NIH) | MH085082 | David J Anderson |
| Howard Hughes Medical Institute (HHMI) | | David J Anderson |
| Jane Coffin Childs Memorial Fund for Medical Research | Postdoctoral Research Fellowship | Prabhat S Kunwar |
| California Institute of Technology | Caltech Divisional Postdoctoral Fellowship | Prabhat S Kunwar |
| National Science Foundation (NSF) | Postdoctoral Research Fellowship | Moriel Zelikowsky |
| National Institutes of Health (NIH) | MH070053 | David J Anderson |
| Simon Foundation | Simon Foundation Collaboration on the Global Brain | Markus Meister, David J Anderson |

The funders had no role in study design, data collection and interpretation, or the decision to submit the work for publication.

## Author contributions
PSK, Conception and design, Acquisition of data, Analysis and interpretation of data, Drafting or revising the article; MZ, Analysis and interpretation of data, Drafting or revising the article; RR, HC, MY, Acquisition of data, Analysis and interpretation of data; MM, Analysis and interpretation of looming data; DJA, Conception and design, Analysis and interpretation of data, Drafting or revising the article

## Ethics
Animal experimentation: This study was performed in accordance with the recommendations in the Guide for the Care and Use of Laboratory Animals of the National Institutes of Health. All of the animals were handled according to approved institutional animal care and use committee (IACUC) protocol 1602, 1552 & 1600.

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
