## [Decision Letter]

Thank you for sending your work entitled “Ventromedial hypothalamic neurons control a defensive emotion state” for consideration at *eLife*. Your article has been favorably evaluated by a Senior editor and 3 reviewers, one of whom, Richard Palmiter, is a member of our Board of Reviewing Editors, and another, Garret Stuber, has agreed to share his identity. A third reviewer remains anonymous.

The Reviewing editor and the other reviewers discussed their comments before we reached this decision, and the Reviewing editor has assembled the following comments to help you prepare a revised submission.

All of the reviewers think that the manuscript provides important behavioral data related to the role of SF1 neurons in mediating innate predator responses and the conceptual framework is appreciated. However, there are a few aspects of the study that need to be addressed before acceptance.

1) The connectivity of the SF1 neurons in terms of both inputs and outputs should be experimentally defined and/or discussed in more detail so that the reader has a better sense of how SF1 neurons are normally activated and how they transmit their activity to other brain regions to modify behavior. The projection to the dPAG, in particular is worth expanding upon.

2) There is some concern that SF1 neuron ablation may have unintended consequences on neighboring neurons. This possibility should be acknowledged and rectified if possible by including or referring to other inhibitory approaches that have been applied to the same neuron population.

3) The authors have promoted their contribution to the intellectual framework of how the SF1 neurons contribute to predator responses without giving appropriate credit to earlier work. A number of specific oversights have been mentioned by one of the reviewers in their minor comments (not shown) that should be acknowledged in a revised version.

---

## [Author Response]

*1) The connectivity of the SF1 neurons in terms of both inputs and outputs should be experimentally defined and/or discussed in more detail so that the reader has a better sense of how SF1 neurons are normally activated and how they transmit their activity to other brain regions to modify behavior. The projection to the dPAG, in particular is worth expanding upon*.

We agree that this is an important issue. To address the issue of the outputs of SF1 neurons, we have now expanded the text and cited earlier neuroanatomic studies that have mapped projections from VMHdm (in the subsection headed “Are emotion states implemented within VMHdm/c itself, or by downstream structures?”). This new discussion highlights the potential role of the dPAG in encoding or mediating emotion states. In this context, we have included a new supplemental figure (Figure 1—figure supplement 1) to illustrate the major projections of SF1^+^ VMH neurons to different brain structures, based on publicly available data in the Allen Brain Mouse Connectivity Atlas database (http://connectivity.brain-map.org), which consists of serial two-photon tomographic reconstructions of Nr5a1-Cre (SF1-Cre) mouse brains injected with a Cre-dependent AAV expressing GFP. We have used the Allen Brain Atlas Brain Explorer 2 program to computationally reconstruct and render in 3D the projections of SF1^+^ neurons to their major targets (Figure 1—figure supplement 1; Videos 9 and 10). This computational analysis, and the raw data on which they are based, are presented with appropriate citations and source attribution, in accordance with the Allen Brain Institute policy and guidelines (http://www.alleninstitute.org/terms-of-use/ and http://www.alleninstitute.org/citation-policy/). All relevant URLs, including direct links to the original source data, are cited in the legend to Figure 1—figure supplement 1.

The inputs to VMHdm/c neurons, which have been established by earlier traditional neuroanatomical studies, are now also described in the Discussion section subtitled “The relationship between VMHdm/c and the amygdala in encoding emotion states,”.

*2) There is some concern that SF1 neuron ablation may have unintended consequences on neighboring neurons. This possibility should be acknowledged and rectified if possible by including or referring to other inhibitory approaches that have been applied to the same neuron population*.

We have now acknowledged and discussed this caveat in the manuscript (in the subsection headed “SF1^+^ neurons control defensive behaviors in multiple contexts”). However this concern is mitigated by the fact that quantitative analysis of Esr1^+^ neurons in the adjacent region of VMH (VMHvl), as well as in the neighboring arcuate nucleus (ARH), revealed no significant reduction in the number of those cells in mice in which SF1^+^ neurons were genetically ablated, compared to control mice (Figure 7). Furthermore, we point out that the effect of this ablation to reduce predator avoidance is similar to the phenotype that was reported by [104], using hM4DREADD as an independent method to silence SF1^+^ neurons. Therefore, we feel confident that the loss-of-function phenotypes we report are indeed due to the loss of SF1^+^ neurons, and not to “collateral damage” to other neighboring cell populations.

*3) The authors have promoted their contribution to the intellectual framework of how the SF1 neurons contribute to predator responses without giving appropriate credit to earlier work. A number of specific oversights have been* mentioned *by one of the reviewers in their minor comments (not shown) that should be acknowledged in a revised version.*

We apologize for these oversights. We have now rectified this issue with text revisions and additional literature citations, which reference earlier studies that provide evidence of an involvement of hypothalamic structures in predator avoidance behaviors and in learned fear. These revisions are incorporated into the Discussion section. We specifically raise the possibility that a target(s) downstream of VMHdm/c, including the dPAG, may act as an emotion center, and cite earlier work germane to this topic as mentioned by the reviewer (in the subsection headed “Are emotion states implemented within VMHdm/c itself, or by downstream structures?).

In this context, we wish to clarify how our findings advance the field beyond earlier studies. Many of the older studies we were asked to cite used pharmacological methods for activation or inhibition that lacked cellular specificity and spatial resolution or, in cases where electrical stimulation or lesioning was used, could not exclude a role for fibers of passage. Moreover, activation and inhibition manipulations were reported in different studies from different laboratories, using different species and different behavioral assays, making direct comparisons difficult. We have performed both optogenetic activation, and targeted ablation, of a genetically defined subset of VMHdm/c neurons, in mice, using an extensive battery of behavioral assays. Our data argue that the function of SF1^+^ neurons is not restricted to predator defense per se, but rather that this population controls defensive behaviors in different contexts containing different types of threats (for example, in anxiety assays). These issues are now discussed in the Discussion section.

Our studies also provide stronger evidence that the hypothalamus encodes emotion states. Previous studies in rodents reached this conclusion by inferring and attributing a state of “fear,” a specific human emotion, to animal models, a practice which has recently been questioned (68), or on verbal report, which can only be assessed in humans. In contrast, our study is the first (to our knowledge) to provide evidence of an emotion state in an animal model using a set of objective and general features, common to different emotions within a species and to similar emotions across species (6), which are independent of anthropocentric homologies and verbal report.

We also provide strong and direct evidence that the hypothalamus can serve as an unconditioned stimulus for emotional learning, clarifying earlier conflicting studies. This issue is particularly important, because it underlies prevailing textbook views of emotion, such as that presented in Principles of Neuroscience (5^th^ Edition, 2013, now cited), which continue to depict the amygdala as the central orchestrator of emotion states, and the hypothalamus as a mere relay or motor output of the amygdala. We now discuss this in detail in the Discussion subsections entitled “SF1^+^ neurons, emotion and emotional learning,” and “The relationship between VMHdm/c and the amygdala in encoding emotion states”.

References mentioned by the reviewer that we have now incorporated into the manuscript are as follows: [54]; [80]; [105]; [119]; [90]; [13]; [5]; [52]; [110]; [62]; Blanchard and Blanchard, 1989; [12].